# Comprehensive evaluation of satellite-based and reanalysis soil moisture products using in situ observations over China

Xiaolu Ling[1,2], Ying Huang*[,3,4], Weidong Guo[3,4], Yixin Wang[3], Chaorong Chen[3], Bo Qiu[3,4], Jun Ge[3,4], Kai Qin[1,2], Yong Xue[1,2], Jian Peng [5,6]

[1] Jiangsu Key Laboratory of Coal-based Greenhouse Gas Control and Utilization, China University of Mining and Technology, 221008, Xuzhou, Jiangsu, China

[2] School of Environment and Spatial Informatics, China University of Mining and Technology, Xuzhou, 221000, China

[3] Institute for Climate and Global Change Research, School of Atmospheric Sciences, Nanjing University, Nanjing, 210023, China

[4] Joint International Research Laboratory of Atmospheric and Earth System Sciences, Nanjing University, 210023, Nanjing, China

[5] Department of Remote Sensing, Helmholtz Centre for Environmental Research−UFZ, Permoserstrasse 15, 04318, Leipzig, Germany

[6] Remote Sensing Centre for Earth System Research, Leipzig University, 04103, Leipzig, Germany

*Correspondence to*: Ying Huang (huangy07@nju.edu.cn)

**Abstract.** Soil moisture (SM) plays a critical role in the water and energy cycles of the earth system; consequently, a long-term SM product with high quality is urgently needed. In this study, five SM products, including one microwave remote sensing product [European Space Agency's Climate Change Initiative (ESA CCI)] and four reanalysis datasets [European Centre for Medium-Range Weather Forecasts (ECMWF) Re-Analysis-Interim (ERAI), National Centers for Environmental

Prediction (NCEP), the Twentieth Century Reanalysis Project from National Oceanic and Atmospheric Administration (NOAA) and the ECMWF Reanalysis 5 (ERA5)], are systematically evaluated using in situ measurements during 1981-2013 in four climate regions at different timescales over the Chinese Mainland. The results show that ESA CCI is closest to the observations in terms of both the spatial distributions and magnitude of the monthly SM. All reanalysis products tend to overestimate soil moisture in all regions but have higher correlations than the remote sensing product except in Northwest

China. The largest inconsistency is found in southern Northeast China region, with an ubRMSE value larger than 0.04. However, all products exhibit certain weaknesses in representing the interannual variation of SM. The largest relative bias of 144.4 % is found for the ERAI SM product under extreme and severe wet conditions in the Northeast China, and the lowest relative bias are found for the ESA CCI SM product with the minimum of 0.48 % under extreme and severe wet conditions in Northwest China. Decomposing mean square errors suggests that the bias terms are the dominant contribution for all products,

and the correlation term is large for ESA CCI. As a result, the ESA CCI SM product is a good option for long-term hydrometeorological applications in the Chinese Mainland. ERA5 is also a promising product especially in North China and Northwest China in terms of low bias and high correlation coefficient. This long-term intercomparison study provides clues for SM product enhancement and further hydrological applications.

## 1 Introduction

Soil moisture (SM) is a key state variable in the climate system and controls the exchange of water, energy, and carbon fluxes between land surface and atmosphere (Western and Blöschl, 1999; Robock et al., 2000; Ochsner et al., 2013; McColl et al., 2017; Peng and Loew, 2017; Qiu et al., 2018). SM can influence runoff generation, drought development and many other processes of hydrology and agriculture (Markewitz et al., 2010; Das et al., 2011; Sevanto et al., 2014; Akbar et al., 2018). Thus, understanding SM characteristics is beneficial to flood prediction (Komma et al., 2008; Norbiato et al., 2008), drought monitoring (Dai et al., 2004; Anderson et al., 2007; AghaKouchak et al., 2015; Li et al., 2018) and water management, which are directly related to crop growth (Engman, 1991; Bastiaanssen et al., 2000; Dobriyal et al., 2012). SM also affects the climate system through land-atmosphere feedback loop (Kim and Hong, 2007; Dirmeyer, 2011; Zuo and Zhang, 2016), while the SM-climate interaction actually amplifies climate variability in some transitional climate zones (Seneviratne et al., 2010). Despite the small total mass of SM compared to other water cycle components, it is essential for numerical weather prediction (An et al., 2016) and has been recognized as an essential climate variable (ECV) (GCOS, 2010).

In situ measurements have been acknowledged as the most accurate method to determine SM values, but they cannot fulfill the demand of high spatial and temporal resolution for hydrometeorological use (Bárdossy and Lehmann, 1998). Furthermore, the temporal coverage of in situ measurements is usually not long enough. Therefore, satellite-based products, reanalysis products and numerical model products are often used (Peng et al., 2017). Although model outputs are spatially and temporally continuous, large uncertainties still exist in model simulations because of the physical structure, parameters, and other reasons (Schellekens et al., 2017). Reanalysis products are generally more accurate, yet they still inherit some uncertainties of the models (Berg et al., 2003), and their spatial resolutions are not high enough for regional application (Crow and Wood, 1999). Despite the short temporal coverage and the limitation of only measuring the surface SM (Petropoulos et al., 2015), satellite-based products are very promising (Chauhan et al., 2003; Bogena et al., 2007; de Jeu et al., 2008) because they are often based on observations with high spatial resolution (Busch et al., 2012). For this reason, satellite-based products are normally taken as reference datasets to evaluate model outputs and reanalysis products (Crow and Ryu, 2009; Lai et al., 2014). To choose the most appropriate SM product for long-term hydrological and meteorological studies, more evaluation work needs to be done. Several evaluation studies have been conducted to find a qualified remote sensing SM product (Li et al., 2009; Zhang et al., 2012; Lai et al., 2014; Peng et al., 2015; An et al., 2016; Ma et al., 2016; Zhu et al., 2018). The SM product from the European Space Agency (ESA) Climate Change Initiative (CCI) program has attracted attention in recent years (Dorigo et al., 2018) and has been proven to have good quality in some regions of the world (Dorigo et al., 2015, 2017; Chakravorty et al., 2016; Ikonen et al., 2018; González-Zamora et al., 2019; Beck et al., 2021). Peng et al. (2015) evaluated the ESA CCI product along with four other datasets in Southwest China and found that it has the potential to provide valuable information. Based on observational data and eight model products, An et al. (2016) further confirmed that the CCI SM can be applied over China. Ma et al. (2016) compared the ESA CCI and ERAI products with in situ measurements and found that both products show

reliable time-series results. However, few studies on long-term SM products over 30 years have been compared with the ESA CCI product using in situ measurements in East China, and thus, more in-depth evaluation needs to be done.

Many efforts have been made to assess the reanalysis products of soil variables based on limited observations (Decker et al., 2012; Hagan et al., 2020). Analysis of spring SM shows that ERAI can well reproduce the interannual variation in observed value and exhibits a better correlation with precipitation and evaporation than NCEP/NCAR R1, Modern-Era Retrospective analysis for Research and Applications (MERRA), Japan Meteorological Agency and the Central Research Institute of Electric Power Industry (JRA) SM products (Liu et al., 2014). Using in situ observations from 25 networks worldwide from 1979 to 2017, ERA5 SM performs better than other reanalysis products, and NCEP products show higher skill in terms of long-term trends (Li et al., 2020). During weak monsoon conditions, ERAI overestimates SM over India, and SM correlates well with observed rainfall (Shrivastava et al., 2017). Using 670 SM stations worldwide, Deng et al. (2020) found that NCEP performed poorly in December-February and June-August and in arid or temperate and dry climates. Nevertheless, to our knowledge, few studies on the estimation of long time series of SM over the Chinese Mainland have been conducted.

The objective of this study is to comprehensively evaluate long-term SM products over the Chinese Mainland and identify the most accurate products for further meteorological and hydrological research. For this purpose, in situ measurements during 1981-2013 are utilized to evaluate five SM products. In addition to the comparison based on different statistical metrics, the source of errors is also discussed.

## 2 Data and Methodology

### 2.1 Remotely Sensed and Reanalysis Products

#### 2.1.1 ESA CCI SM

Generated by the ESA Program - Climate Change Initiative CCI project (ESA CCI), ESA CCI SM includes active, passive and combined products (Liu et al., 2012; Gruber et al., 2017). The ESA CCI SM v04.4 combined product is employed in this study, which provides SM data starting from November 1978 until June 2018 with a spatial resolution of 0.25°. The project of ESA CCI is to use C-band microwave scatterometers (Aqua satellite and the Advance Scatterometer, ASCAT) and multi-channel microwave radiometers (SMMR, SSM/I, TMI, AMSR-E, WindSat, AMSR2) to produce a long-term reliable time series of SM (Chakravorty et al., 2016). The ESA CCI SM v4 is better at detecting SM changes (Balenzano et al., 2011) than previous versions, as it merges all active and passive Level 2 products directly to generate the combined product, rather than creating active and passive products separately and then merging together (ESA, 2018; Gruber et al., 2019). Global Land Data Assimilation System Noah (GLDAS 2.1) was used as a scaling reference in the combined product to obtain a consistent climatology, flagging of high vegetation optical depth (VOD) for Soil Moisture and Ocean Salinity (ESA SMOS) and AMSR-2 method changed (Dorigo et al., 2017; Pasik et al., 2020). A polynomial SNR-VOD regression and the p-value based mask was used to fill spatial gaps in TC-based SNR estimates, and exclude unreliable input dataset in the combined product,

respectively. Here, we evaluate all the products over the period from 1981 to 2013 (the same as below), during which in situ measurements are also available. The top layer of ESA CCI SM data at the depth of 2-5 cm depth are estimated.

### 2.1.2 ERAI SM

ERAI is a famous reanalysis product produced by the European Centre for Medium-Range Weather Forecasts (ECMWF, 2009). The data assimilation system is based on the Integrated Forecast System (IFS Cy31r2), which includes a four-dimensional analysis with a 12-hour analysis window. The ERAI data used in this study are on a fixed grid of 80 km and have a temporal resolution of four times daily and monthly. ERAI starts in 1979 and is continuously updated in real time (Paul et al., 2011). ECMWF simulates SM at 4 depths: 0-7 cm, 7-28 cm, 28-100 cm, and 100-255 cm. As suggested by An et al. (2016), the data at depths of 7-28 cm are linearly interpolated to a depth of 10 cm for evaluation.

### 2.1.3 NCEP SM

NCEP is the 2nd reanalysis product provided by the National Centers for Environmental Prediction and Department of Energy (NCEP-DOE, (Kanamitsu et al., 2002)). The product is available since Jan 1979 with a spatial resolution of approximately 200 km. The temporal resolution includes 4 times daily and monthly data. NCEP has two layers of SM between 0-10 cm and 10-200 cm, in which the first layer was chosen for evaluation.

### 2.1.4 NOAA SM

The Twentieth Century Reanalysis Project (20CR) led by the Earth System Research Laboratory Physical Sciences Division from the National Oceanic and Atmospheric Administration (NOAA) and the University of Colorado Cooperative Institute for Research in Environmental Sciences (CIRES) also produces a long-term SM product. The version of V2c is used here, spanning the entire twentieth century from 1851 to 2014 (Compo et al., 2011). The NOAA SM product is generated with a spatial resolution of 2 degrees at six hours (also monthly) and with 4 subsurface levels (0, 10, 40, 100 cm), among which the data at 10 cm depth are used.

### 2.1.5 ERA5 SM

ERA5 is the latest reanalysis product produced by ECMWF, covering the period from 1979 to present. The product uses a new version of the ECMWF assimilation system IFS (IFS Cycle 41R2), and combines vast amounts of historical observations, including ozone, aircraft and surface pressure data, as well as various newly reprocessed datasets and recent instruments that could not be ingested in ERA-Interim (C3S, 2017). The ERA5 model input includes the World Climate Research Programme (WCRP) Coupled Model Intercomparison Project (CMIP) greenhouse gases, volcanic eruptions, sea surface temperature (SST), and sea-ice cover, which are appropriate for climate. Furthermore, the spatial (31 km globally) and temporal (hourly) resolutions of ERA5 are rather high compared to ERAI. ERA5 will eventually cover the period from 1950 to the present, and one of its key improvements is better SM (Komma et al., 2008). The land surface models of the Interactions between Soil,

Biosphere, and Atmosphere (ISBA) driven by ERA5 also show consistent improvements, especially in surface SM, compared to those driven by ERAI (Albergel et al., 2018). Similar to ERAI, ERA5 also has 4 levels of SM data, in which the SM is interpolated to 10 cm for evaluation.

### 2.2 In Situ SM and Preprocessing of Datasets

The in situ SM observations were generated by three SM datasets as follows:

    (1)  The International Soil Moisture Network (ISMN)

The updated Chinese soil moisture was presented as volumetric soil moisture ($\theta_v$, unit=m$^3$ m$^{-3}$) for 1981 to 1999 from the International Soil Moisture Network website (https://ismn.geo.tuwien.ac.at/en/) (Dorigo et al., 2011). The ISMN provides a global in-situ soil moisture database, which has been widely used for validation of satellite products and model simulation (e.g. Albergel et al., 2012). The SM data at the depth of 0-5 cm and 5-10 cm was obtained and averaged as the value at the depth of 0-10 cm.

    (2)  Soil water content from agricultural-meteorological stations

The in situ SM measurements are obtained from the National Meteorological Information Center of China (NMIC, 2006). The data were collected at 778 agricultural-meteorological stations with a temporal resolution of 10 days since May 1991 (on 8$^{th}$, 18$^{th}$, and 28$^{th}$ each month). As there are too many missing observations after 2013, the evaluations of the different datasets are performed until December 2013. The SM data was observed at the depth of 10 cm, 20 cm, 50 cm, 70 cm, and 100 cm using drying methods, with the data at 10-cm depth utilized. In addition, the observed SM is expressed as the relative water content ($\theta'$, unit=%), while the SM in all other products is in the unit of volumetric water content ($\theta_v$, unit=m$^3$ m$^{-3}$). Therefore, the observed SM is calculated by:

$$\theta_v = \theta' \times \theta_f \times \rho_b / \rho_w \qquad (1)$$

where $\theta_f$ is the field capacity, $\rho_b$ is the dry bulk density, and $\rho_w$ is the water density with a value of 1.0 (unit=g cm$^{-3}$).

    (3)  Mass percent of measured SM

Another dataset including SM, field capacity and dry bulk density in China was recorded from 1981 to 1998, were obtained from the National Meteorological Information Center of the China Meteorological Administration. SM was presented as a mass percentage three times each month to avoid auxiliary calibration (Robock et al., 2000). The volumetric soil moisture is calculated by:

$$\theta_v = \theta_m \times \rho_b / \rho_w \qquad (2)$$

in which $\theta_m$ is the mass percent of measured soil moisture. Within a certain period, the two parameters of $\theta_f$ and $\rho_b$ can be treated as constant. The SM mass percent was measured at 11 levels including the depth of 0-5 cm, 5-10 cm, 10-20 cm, 20-30 cm, 30-40 cm, 40-50 cm, 50-60 cm, 60-70 cm, 70-80 cm, 80-90 cm, and 90-100 cm. To match other datasets, the values at 10 cm depth are calculated by averaging the values at the depth of 5-10 cm and 10-20 cm.

Considering that the field capacity and the dry bulk density are not measured at all stations, data from 119 stations are selected from 1981 to 2013. Not all in situ data were suitable for evaluation given instrumental error and observational conditions, for example, the available measurement period, installation depth and sensor placement. Therefore the evaluation was conducted in unfrozen and snow-free seasons, such as June-July-August (JJA). The selection of appropriate SM values is based on quality control by removing abnormal data due to instrument failures and threshold control by retaining the value between 0-1. First, if there were multiple data points in the same time period, the ISMN SM value was selected if available, or the average of the remaining two datasets was calculated. Second, SM values greater than 3 times the standard deviation were deleted. On considering the availability, all the in situ observations were averaged to monthly data at a depth of 10 cm. The distributions of the available stations are presented in Fig. 1, and detailed information of all the above SM products is listed in Table 1.

## 2.3 Land Surface Air Temperature, Precipitation, and Radiation

The land surface air temperature and precipitation data are obtained from the National Meteorological Information Center (NMIC) at a spatial resolution of 0.25° spanning from 1961 to the latest (http://data.cma.cn/site/index.html). By interpolating from Chinese ground high density stations (over 2400 observation stations), the station observational meteorology dataset (CN05.1) includes daily mean temperature, maximum/minimum temperature and precipitation (Wu and Gao, 2013). The net radiation data were downloaded from the ECMWF ERA5 products, for which the detail information can be referred in section 2.1.5.

The self-calibrating Palmer drought severity index (SC-PDSI) was utilized to determine the performance of all products under different drought/wet conditions (Wells et al., 2004). By adjusting the climatic characteristics and calculating the duration factors based on the characteristics of the climate at a given location, the SC-PDSI has been widely used in recent decades. The SC-PDSI fit Palmer's 11 categories to allow for comparisons across time and space. A negative value indicates drought conditions, and a positive value indicates a wet spell. The source code to the SC-PDSI can be downloaded via the National Agricultural Decision Support System (NADSS; online at http://nadss.unl.edu/).

## 2.4 Evaluation Strategies

### 2.4.1 Statistical metrics

The comparisons were conducted through the statistical metrics, such as the Bias, relative Bias (rBias), Pearson correlation coefficient (R), root mean square difference (RMSD), and the unbiased root mean square error (ubRMSE) using the following formulas:

$$Bias = \frac{\sum_{t=1}^{n}(x_{p,t} - x_{obs,t})}{n} \qquad (3)$$

$$rBias = \frac{Bias}{Mean(Observation)} \qquad (4)$$

$$R = \frac{\sum_{t=1}^{n}(x_{obs,t}-\mu_{obs})(x_{p,t}-\mu_p)}{\sqrt{\sum_{t=1}^{n}(x_{obs,t}-\mu_{obs})^2}\sqrt{\sum_{i=1}^{n}(x_{p,t}-\mu_p)^2}} \tag{5}$$

$$RMSD = \sqrt{\frac{\sum_{t=1}^{n}(x_{p,t}-x_{obs,t})^2}{n}} \tag{6}$$

$$ubRMSE = \sqrt{RMSD^2 - Bias^2} \tag{7}$$

in which n is the total number of time steps, $x_{p,t}$ and $x_{obs,t}$ is the value of SM products (including remote sensing and reanalysis) and observation at time-step t, $\mu_{obs}$ and $\mu_p$ are the mean of the in situ observed values and all SM products, Mean(observation) is the average of observation. The metrics of rBias was used to study the performance of various regions under different drought or wet conditions. The ubRMSE is introduced to evaluate temporal dynamic variability to get rid of the bias error caused by the mismatch of spatial representativeness between the in situ data and all SM products (Jackson et

al., 2010, 2012; Entekhabi et al., 2014). What is worthy to say, the in situ observation were not considered as 'true' value because of instrumental errors and representativeness, so the RMSD terminology was used in this study.

### 2.4.2 Decomposition of mean square errors (MSEs)

To better explain the disagreement between all the SM products and in situ observations, the mean square errors (MSEs, as defined in Eq.(8)) of each product in individual regions are utilized. To decompose the MSEs, the Nash-Sutcliffe efficiency

(NSE, Nash and Sutcliffe, 1970) are utilized as defined in Eq.(9).

$$MSE = \frac{1}{n}\sum_{t=1}^{n}(x_{p,t} - x_{obs,t})^2 \tag{8}$$

$$NSE = 1 - \frac{\sum_{t=1}^{n}(x_{p,t}-x_{obs,t})^2}{\sum_{t=1}^{n}(x_{obs,t}-\mu_{obs})^2} = 1 - \frac{MSE}{\sigma_{obs}^2} \tag{9}$$

NSE was decomposed as the correlation, the conditional bias, and the unconditional bias as showed in Eq.(9) (Murphy, 1988).

$$NSE = A - B - C \tag{10}$$

$$A = R^2$$

$$B = [R - (\sigma_p/\sigma_{obs})]^2$$

$$C = [(\mu_p - \mu_{obs})/\sigma_{obs}]^2$$

in which R is the correlation coefficient of observations and products, $\sigma_{obs}$ and $\sigma_p$ are the standard deviation of in situ data

and all SM products. The Eq.(10) can be transformed as Eq.(11), representing the correlation, the bias and the variability.

$$NSE = 2 \cdot \alpha \cdot R - \alpha^2 - \beta_n^2 \tag{11}$$

$$\alpha = \sigma_p/\sigma_{obs}$$

$$\beta = (\mu_p - \mu_{obs})/\sigma_{obs}$$

Finally, the Eq.(12) was obtained by substituting Eq.(11) into Eq.(9) as follows:

$$MSE = 2\sigma_p\sigma_{obs}(1 - R) + (\sigma_p - \sigma_{obs})^2 + (\mu_p - \mu_{obs})^2 \tag{12}$$

The MSE was decomposed to quantify the contributions of the correlation term, standard deviation term and bias term (Gupta et al., 2009). On the right-hand side of the equation, the first term (correlation term) shows the correspondence between the SM product and the in situ observations. The second term (standard deviation term) explains the degree of similarity of variations, and the third term (bias term) shows the accuracy of the product. With a better understanding of the error structure of the datasets, we can well explain the discrepancy between the SM products and the in situ observations (Dorigo et al., 2010).

## 2.5 Study Area

China is located on the eastern coast of Asia, immediately to the west of the Pacific Ocean. It extends roughly from 3.5°N to 53.75°N latitude and from 73.25°E to 135.25°E longitude. Considering climate conditions and the distribution of available SM data, all estimations are conducted in four research regions as suggested by (Ma et al., 2016), which are shown in Fig. 1. Detailed information on the four research regions is specified in Table 2. Figure 1 also shows annual mean precipitation data obtained from 160 Chinese meteorological stations during 1971-2000 from the National Climate Center of China (NCCA). The 30-year averaged annual mean precipitation is treated as the climatological mean precipitation to define the division of the climate zone.

The comparisons were performed as follows: (i) make a correspondence between all soil moisture data sets and in situ SM by using the values at the nearest neighbor grids; (ii) compare all the SM products at regional scales by calculating the reginal average of monthly value of all SM products, which has been proved can reduce the uncertainty caused by grid mismatch to some extent (Nie et al., 2008); (iii) if the in situ observation were missing, all reanalysis data at the same period were also treated as missing value, which were not taking into account.

## 3 Results and Discussion

### 3.1 Spatial Pattern of SM

Figure 2 shows the spatial patterns of the 33-year averaged SM for the in situ observations and five products. ESA CCI has the highest spatial resolution, followed by ERAI and ERA5, and the spatial resolutions of NCEP and NOAA products are relatively coarse. Considering the frozen and vegetation cover, only the JJA SM values are used for evaluation of spatial pattern. Generally, most SM products are able to capture the overall spatial distribution of the SM value, although the NOAA SM is highly overestimated all through the region. According to the in situ observations, SM is the lowest in the northwest and increases to the northeast and southeast. Except for NCEP, all the other datasets are able to represent the wet center in the northeast of China. ESA CCI underestimates SM in northern Northeast China and in Northwest and Southwest China. SM is underestimated by ESA CCI, but overestimated for all the analysis datasets, except in Northwest China. For the ERA5 dataset, the region in the north of Northwest China is much drier than the other products, with average value less than 0.05 m$^3$ m$^{-3}$. ERAI and ERA5 SM products are able to represent the decreasing trend from southeast to northwest, which is failed for the

NCEP SM. The largest biases reaching 0.15 $m^3$ $m^{-3}$ are found in southern Northeast China, and the largest inconsistency is found in the northwest.

The distribution of the ubRMSE for all stations is shown in Fig. 3 to evaluate temporal SM dynamical variability. By removing the bias, the NCEP product has the lowest ubRMSE with values between 0.01 and 0.03 $m^3$ $m^{-3}$, indicating its better performance at capturing the temporal variation of in situ SM. Large ubRMSE are found for the ESA CCI with values large than 0.04 $m^3$ $m^{-3}$, indicating that this remote sensing product needs to be improved at temporal variation. Spatially large ubRMSE are also found in the Yangtze-Huai region and in the south of Northeast China, which may be attributed to the high SM values. A possible explanation for poor performance in the NC region might be that this region is strongly influenced by irrigation.

## 3.2 Temporal Variability of SM

As referred in Table 2, all temporal variabilities of SM are averaged over the Northeast China, North China, Yangtze-Huai, and Northwest China regions, which are abbreviated as NE, NC, YH, and NW, respectively, below.

### 3.2.1 Temporal Evolution

The temporal evolutions of in situ observations and grid point SM values from the five datasets are averaged over each research region during JJA, as displayed in Fig.4. Generally, all the reanalysis products have positive bias of 0.08-0.15 $m^3$ $m^{-3}$, 0.05-0.10 $m^3$ $m^{-3}$, 0.07-0.13 $m^3$ $m^{-3}$, and 0.01-0.05 $m^3$ $m^{-3}$ in the NE, NC, YH, and NW regions, respectively. ESA CCI tend to have negative bias with observations around -0.06-0 $m^3$ $m^{-3}$. All products perform well in the NW region, and the worst performance is found in the NC region. ERAI largely overestimates SM in all the research regions, while NOAA and NCEP SM has the lowest bias among the reanalysis datasets. Reanalysis can better reproduce the variation characteristics than remote sensing during extreme events period, probably due to large percent of missing data, and instrument constrict.

Table 3 shows the biases, RMSD, ubRMSE and correlation coefficients for the comparison between all products and in situ observations during 1981 to 2013. All the evaluation indexes were calculated using monthly spatial average over all regions. ESA CCI presents the lowest biases for all regions, indicating that ESA CCI is the closest to the observed SM values. ERAI SM has the largest positive bias for all regions. By removing bias error, the ubRMSE for all products fluctuate between 0.016 and 0.025 except for the NC region, indicating poor performance in capturing the temporal variability. The correlation coefficient with observations for ESA CCI is relatively low. Good correlation is obtained for ERA5 SM except in the NC region, indicating that ERA5 can well represent the temporal and spatial variation. All products show small correlation in the NC and NW regions, implying that none of the products can capture the spatial-temporal variation of SM over both regions. The Taylor diagrams presenting the statistics of the comparison between ESA CCI, NCEP, ERAI, NOAA, ERA5 and in situ observations over four regions are shown in Fig. 5. Generally, the NOAA SM is highly overestimated in all regions, and ESA CCI SM is underestimated. Most correlation coefficient values are between 0.5 and 0.6 for ERA5, implying a good performance of variability. Lower correlations are found for ESA CCI and ERAI SM, demonstrating that both products represent poor performance of changing characteristics. All products exhibit poor correlations in the NW region.

### 3.2.2 Seasonality

Monthly SM from 1981-2013 during unfrozen and snow-free months have also been calculated in Figure 6, showing temporal evolution of SM seasonality averaged spatially over different regions. Overall there exists a negative and a positive bias between remote sensing and reanalysis with respect to SM observations, respectively. The difference in ESA CCI is smaller than all reanalysis products, especially in the period where in situ SM value is low, which is in line with Ma et al. (2019) that ESA CCI have relative poor skills with lower time series correlations in sparse or dense VOD conditions but good performance in moderate densely vegetated areas (Zeng et al., 2015). Furthermore, soil types (silt, clay, sand) also plays an important role in terms of different regions. Chakravorty et al. (2016) studied the influence of soil texture on regional scale performance and found that large fractional RMSE is associated with large percentage of sand, might be one of the reasons that poor performance is found in the NW region. ESA CCI yields the worst seasonal cycle results with respect to temporal variation, which may be because of large percentage of missing data. Furthermore, the remote sensing products are completely independent without assimilating or integrating measured observations. Seasonal cycle of SM in the NE region is obvious, partly due to the sufficient water content there. Observed SM in all regions reaches its minimum from April to June, and then increases to its maximum from July to September, which can be reproduced by all reanalysis. All reanalysis SM series have a larger dynamic range than in situ observations and remote sensing SM values. ERA5 are closer to the observations in the NC and NW regions, while NCEP and NOAA show the smallest biases in the NE and YH regions. ERA5 SM performed better than ERAI as show a similar variation tendency with the observations and smaller difference, with the average relative biases of 7.40 %, 18.70 %, 7.34 %, and 15.38 % in the NE, NC, YH, and NW regions, respectively.

Figure 7 displays the autocorrelation coefficients lagging one month in different seasons to investigate the persistence of the soil moisture anomaly for in situ observations and five products. The aim of this figure is to study the soil moisture memory in different seasons. It is shown from observations that the autocorrelation is high in spring and autumn, indicating the soil moisture are obviously affected by the value one month before in spring and autumn. Autocorrelation is low in summer and winter, implying that SM in these seasons are strongly affected by meteorological elements, such as the influences of liquid and solid precipitation and frozen. The ESA CCI correlation are low during MAM, JJA, and SON seasons because of the large amount of missing data. The lowest autocorrelation coefficient is found in the NW region, possibly because of the particular sand soil with relative high porosity and low water holding capacity. The regions with good persistence of soil anomalies are located in the northwest and eastern northeast regions of China, which are dominated by relatively simple land cover, for example, bare soil and forests, respectively. The NOAA SM shows larger autocorrelations for all seasons than the other reanalysis products, implying that NOAA models should take into account the influence of some other variables on soil moisture in the future, for example, temperature and precipitation. ERA5 shows better performance than ERAI, especially with a close autocorrelation coefficient in the NE region. The information of soil moisture autocorrelation gives hint for the assimilation of surface soil moisture into land surface models (Crow and Van den Berg, 2010), in which during summer and winter, the influence of meteorological elements (e.g., precipitation, temperature, evaporation, etc) should be considered more.

### 3.2.3 Interannual Anomalies

JJA SM shows evident interannual anomalies in all the research regions, as shown in Fig. 8. Most peaks and troughs can be well represented by all products in the NE and YH regions, while the variation characteristics cannot be reproduced in the other regions, especially in the NC region. Furthermore, all products have a smaller amplitude of variation than observations in extreme wet or drought years in the NC and NW regions, implying that models showed poor ability in representing extreme events.

Specifically, the variation range of the NOAA SM is the largest especially in wet and drought years in the NE region. Taking the years of drought from 2001 to 2002 and the wet year of 2003 as examples, this characteristic was missed by ESA CCI. The variation range of NCEP SM is significantly smaller than the actual measurement, and the simulation of NCEP is obviously inferior to the other three products. In the NC region, all products fail to capture the JJA SM variation tendency, especially during extreme drought and wet periods. NOAA and ERA5 can capture the basic trend, but the variation range does not match the measured value. The variation amplitudes of NCEP and ERAI are obviously smaller than the observations. Surface SM is a variable associate with precipitation and evaporation, both of which fluctuate greatly with time in the JJA seasons. To improve the quality of SM, all reanalysis data would improve their performance in representing precipitation and evaporation, especially during extreme events. In the YH region, ERAI and ERA5 can roughly reproduce the trend of change, but the magnitude of the change is large. There is a SM peak occurring in 1998, in accordance with the 1998 Massive Flood. The peaks in the year of 1987, 1998, and 2001 can be reproduced by the all products. In the NW region, none of these products are able to reproduce the variation characteristics, especially with worse performance in drought periods than in wet periods. According to the correlation (in Table 3), ERA5 has the best performance, but it shows a fictitious increase from 1981 to 1993.

### 3.3 Decomposition of the Mean Square Error (MSE)

The (a) contribution to MSE is decomposed into a correlation term, standard deviation term and bias term according to Eq. (12) and (b) their fractions are showed in Fig. 9. The contribution of the bias term is much larger than the correlation term except for the ESA CCI and ERA5 in the NW region, indicating that reducing biases is the direction we need to follow to further improve the quality of reanalysis SM products. The MSE of ESA CCI SM is the smallest for all regions with a large fraction of correlation term, indicating that the main error of ESA CCI comes from the poor performance of the variation tendency. The MSE of ERA5 performs inconsistently that its main difference comes from the correlation term in the NC and NW regions, while the bias terms are dominant in the NE and YH regions. This implies that improving spatio-temporal resolution and assimilating more observation might be a potential way to improve SM estimate, but the large fraction of ERA5 also point to the need for improving model simulation ability of SM. Additionally, all products present poor performance in the NC and NW regions with a high correlation term. The standard deviation term has little effect on MSE for all datasets except for the ESA CCI in the NE region and ERAI product in the NC region. The NOAA SM product also shows a small

MSE except in the NW regions, which is similar to previous evaluations in some other regions (Peng et al., 2015; An et al., 2016; Zhu et al., 2018).

## 3.4 SM Performance Under Various Climate Background

Figure 10 shows the rBias under different humid/arid conditions by utilizing SC-PDSI (Wells et al., 2004). The rBias of JJA SM between in situ observation and remote sensing/reanalysis was calculated at each in situ grid point as the bias divided by the mean of in situ observations, and then averaged over regions. All of the reanalysis products show a lower rBias under drought condition than wet condition, indicating better performance of all products under dry conditions. The largest rBias was found for all products in the NE region, implying that the largest uncertainty would appear in this region during extreme

events. Large difference of rBias between dry and wet conditions was observed in the NW region, implying that all products fail to represent the SM value when the water content is high. The largest rBias is found for ERAI under severe wet conditions in NE, with an average bias of 144.4 %. The best performance is found for ESA CCI SM in NW, with averaged rbias of 10.0 %, respectively.

For the ubRMSE in different regions (Fig. 11), the ubRMSE of all SM products in the NE and NW regions is noticeably high.

The difference of ubRMSE between different conditions are not so large as rBias, especially in the NE region. Overall the ubRMSE for all products is larger under wet conditions, while the phase is opposite in the NW region. The averaged bias for ESA CCI under drought conditions is smaller than that under wet conditions. The largest and smallest ubRMSE are found for the ESA CCI under wet condition in the NE region and NCEP SM products under both conditions in the YH region, respectively.

Previous studies have shown that soil moisture is influenced by the combination of precipitation and evaporation, in which land surface evaporation is linked with temperature and surface net radiation (Jasper et al., 2006; Harmsen et al., 2009). Figure 12 shows scatter plots of (a, d, g) precipitation, (b, e, h) temperature, and (c, f, i) net radiation anomalies versus observed SM anomalies over different regions in (left column) MAM, (middle column) JJA, and (right column) SON seasons. Obvious positive correlations are found between precipitation and SM in the YH regions during MAM and SON seasons, and in the

NE and NC regions during JJA season. Temperature and net radiation show negative correlation with in the NE, NC, and YH regions. The correlation coefficient is low for all meteorological variables in the NW region, which may be attributed to the large fraction of sand there. Soil moisture in the NE and NC regions tends to be influenced by temperature during cold seasons. SM in the YH region tend to be influenced by radiation during warm seasons, due to the large evaporation there.

## 3.5 Discussion

ESA CCI SM product showed the top layer soil content at 5-cm depth or so. The in-situ measurement depth and model output are at the depth of 0-10cm, which were also treated as the top layer soil content. Such difference would also cause representativeness errors. Previous studies have found that there is a close relationship between surface SM and SM in the upper ten centimetres (i.e., Albergel et al., 2008; Dorigo et al., 2015), so the SM measurements at the depth of 10 cm were

chosen as the reference to evaluate satellite-based and reanalysis products. Furthermore, introducing ubRMSE and conducting comparison at regional scale can remove the bias error caused by mismatch of grid cell to some extent.

The ESA CCI combined data generally increase the number of observations available for a time period but the correlation coefficients were not better than those of the best performing single dataset (Dorigo et al., 2015). Dorigo et al. also studied the possible reasons of input data, and found that the low correlation of combined product possibly due to the merging procedure, including the influence of vegetation (Taylor et al., 2012), the different original overpass time, and the scaling of high resolution ASCAT product to lower resolution reference products. Beck et al. (2021) found that ESA CCI SM performed better in eastern Europe in terms of high-frequency fluctuations, and found that the overall performance of ESA CCI may be not so good was possibly due to the incorporation of ASCAT that performed less well. Furthermore, the poor correlation of remote sensing product is also associated with missing of available data because of instrument constrict and cloud impact.

In the winter, SM decreases in all regions mainly because of decreased precipitation. Lower evaporation caused by sudden cooling may explain why SM increases in early winter. SM reaches a local minimum in the spring in most of the regions except the NE region, as a temperature rise leads to higher evaporation, while precipitation does not increase much in this season. In the NE region, ice and snow melting partially compensates for soil water loss and helps maintain a relatively stable SM. Increased precipitation in the summer gives rise to an evident increase in SM. In the autumn, SM continues to increase in the YH and NC regions, probably due to less evaporation caused by lower temperatures.

Precipitation and evaporation are found to be the most important determinant of soil moisture simulation performance, in which the evaporation is associated with temperature and radiation (Gottschalck et al., 2005; Mall et al., 2006; Chen & Yuan, 2020). SM value in the analysis is overestimated, partly due to the reason that the JJA precipitation over China is overestimated by models (e.g., Luo et al., 2013; Yun et al., 2020). The largest bias of precipitation overestimation using the hourly 31-km-resolution ERA5 reanalysis data is found over the Tibetan and Yun-Gui Plateaus, the North China Plain, and the southern mountains, which gives one of the explanations why reanalysis products represent the worst performance over the NC region. Soil type and soil texture are also important elements for soil moisture estimation. In the southwest of the NE region, the sand fraction of the topsoil can reach about 80 %-90 %, and the sand fraction and clay fraction of the topsoil are around 30 %-40 % and 10 %-30 % respectively (Shangguan et al., 2012) in the north NE region. The inconsistent of the soil types over the NE region might interpret why the large inconsistency of spatial distribution were found for all products. In the northwest of the NW region, sand fraction is larger than 80 %, and the sand fraction is low in the southeast of the NW region. The large difference of soil types over the north NW region is one of the reasons that all products show poor performance. In the NC and YH regions, sand and clay fraction of the topsoil account for about 10 %-20 % and 30 %-50 %, 30 %-50 % and 0-20 % respectively. The different performance over the NC and YH regions give hints that remote sensing and reanalysis products tend to performance worse when the soil type is sand because of its poor water retention.

ERA5 (~0.28125°) has a higher spatial resolution than ERAI (~0.75°), which can be directly reflected in their spatial patterns of SM distribution. ERA5 can well reproduce the spatial distribution and time series of monthly SM over the Chinese Mainland in terms of low bias between observations. Looking at the monthly variation and interannual variation in the SM anomaly,

ERA5 has better performance than ERAI in terms of low bias. It is proposed that ERA5 will eventually replace ERAI, and we do see improvements in the ERA5 product. However, ERA5 overcorrects the problem of small variation in ERAI, which leads to almost the same ubRMSE and correlation coefficient in ERA5 and ERAI. This imply that only improving model resolution and assimilating satellite SM estimates can help reduce the difference of SM, but not improve much in the spatial and temporal variation at long-term scales. This might be caused by the little improvement of assimilating the ASCAT soil moisture in the ERA5 reanalysis (Hersbach et al., 2020). Beck et al. (2021) concluded that assimilating satellite soil moisture estimate may not improve more than increasing model resolution or improving soil moisture simulation ability, which is in line with our results. This suggest that improving model simulation performance of SM is beneficial especially at long-term scales.

## 4 Conclusions

To evaluate the performance of long-term SM products over the Chinese Mainland, one satellite-based product and four reanalysis datasets from 1981 to 2013 are selected for comparison with in situ measurements at different time scales.

Overall, ESA CCI has the best performance with the highest spatial resolution and accuracy, making it a good option for long-term hydrometeorological applications in China. The 0.25°*0.25° resolution of the ESA CCI product produces the finest spatial pattern of SM, making it more beneficial for regional application than other SM products. However, ESA CCI shows poor performance in terms of its low correlation and missing values, especially in Northeast China.

ERAI and ERA5 can well reproduce the tendency of the time series and perform best at stations, but they overestimate the seasonal variation in SM. ERA5 is also a promising product with better performance in several aspects than ERAI, highlighting the importance of incorporating more observations at finer spatial resolution.

NCEP cannot reproduce the spatial pattern of SM in China, the time series of NCEP SM data is poorly correlated with observations, and the variation amplitude of its seasonal cycle is much larger than that of the observations. NOAA is able to reproduce the basic spatial pattern, but it systematically overestimates SM in China and shows little seasonal variation. All the SM products used in the present study cannot adequately simulate the interannual variation in the SM anomaly.

The mismatch between SM layers in analysis products and observations, as well as their spatial mismatch, should be investigated in the future (Choi and Hur, 2012; Crow et al., 2012). Furthermore, sub-daily SM model products considering the advantages of individual models under different weather regimes and climate scenarios would be merged in future work (Chen and Yuan, 2020).

***Data Availability.*** We acknowledge the data providers of the following SM products: The updated Chinese soil moisture presented as volumetric soil moisture ($\boldsymbol{\theta_v}$, unit=m$^3$ m$^{-3}$) for 1981 to 1999 was downloaded from the International Soil Moisture Network website (https://ismn.geo.tuwien.ac.at/en/). The in situ SM measurements are obtained by requesting from the website of National Meteorological Information Center of China (NMIC, http://data.cma.cn/site/index.html). ESA CCI (http://www.esa-soilmoisture-cci.org), ECMWF ERAI (https://apps.ecmwf.int/datasets/data/interim-full-daily/levtype=sfc/), ERA5 (https://apps.ecmwf.int/data-catalogues/era5/?class=ea), NCEP (doi:10.1175/BAMS-83-11-1631), NOAA (doi:10.1002/qj.776) and NMIC (http://data.cma.cn/data/cdcdetail/dataCode/AGME_AB2_CHN_TEN.html). The land

surface air temperature and precipitation data are obtained from the National Meteorological Information Center (NMIC) at a spatial resolution of 0.25° spanning from 1961 to the latest (http://data.cma.cn/site/index.html). The source code to the SC-PDSI can be downloaded via the National Agricultural Decision Support System (NADSS; online at http://nadss.unl.edu/).

*Author Contribution.* YH, WG, and JP designed the study and performed the experiments; XL and YW performed the experiments, analyzed the data, wrote and revise the manuscript, BQ, JG, KQ, and YX contributed to the interpretation of the results and revision of the manuscript.

*Competing Interests.* The authors declare that they have no conflict of interest.

*Acknowledgements.* This work was jointly supported by the National Key R&D Program of China (2017YFA0603803), National Science Foundation of China (Grant No. 42075114, 41705101, 41775075), Priority Academic Program Development of Jiangsu Higher Education Institutions (140119001), and ESA MOST Dragon 5 programme (Monitoring and Modelling Climate Change in Water, Energy and Carbon Cycles in the Pan-Third Pole Environment, CLIMATE-Pan-TPE).

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

**Figure Captions**

Figure 1. Spatial distribution of 119 agricultural-meteorological observation stations and four research regions over China for the period 1981 to 2013. The colors denote the distribution of annual precipitation (unit: mm a$^{-1}$) from 1971-2000.

Figure 2. Annual averages of (a) observations and (b-e) five satellite/reanalysis SM products (units: m$^3 \cdot$m$^{-3}$) during June to August for the period of 1981 to 2013 in China.

Figure 3. Same as Figure 2 but for ubRMSE.

Figure 4. Time series of SM in four research regions (a-d) from 1981 to 2013.

Figure 5. Taylor diagrams of the comparison between multisource SM products and in situ observations. Ref. is the SM from in situ observations.

Figure 6. Seasonality of SM distributions based on in situ observations and five products averaged over the (a) NE, (b) NC, (c) YH, and (d) NW regions from 1981 to 2013.

Figure 7. Distribution of autocorrelation coefficient of SM in seasons: (1st row) FMA and MAM; (2nd row) MJJ and JJA; (3rd row) ASO and SON; (4th row) NDJ and DJF.

Figure 8. Temporal evolution of the JJA SM anomaly time series from observations and the five satellite/reanalysis products in four research regions from 1981 to 2013.

Figure 9. The (a) decomposition of three terms to the mean square errors (MSEs) for the four satellite/reanalysis products from 1981 to 2013, as well as (b) their fraction.

Figure 10. The relative bias of remote sensing and reanalysis SM against in situ observations under dry or wet conditions in different regions.

Figure 11. The ubRMSE of remote sensing and reanalysis SM against in situ observations under dry or wet conditions in different regions.

Figure 12. Scatterplots of monthly anomalies of (a, d, g) precipitation, (b, e, h) temperature, and (c, f, i) net radiation vs observed soil moisture in the top 10 cm depth during 1981-2013 during (a, b, c) MAM, (d, e, f) JJA and (g, h, i) SON seasons. R is the correlation coefficient over four research regions, and the values marked with *, **, and *** means that the correlation coefficient has passed the significance test of 90 %, 95 % and 99 %, respectively.

**Figures**

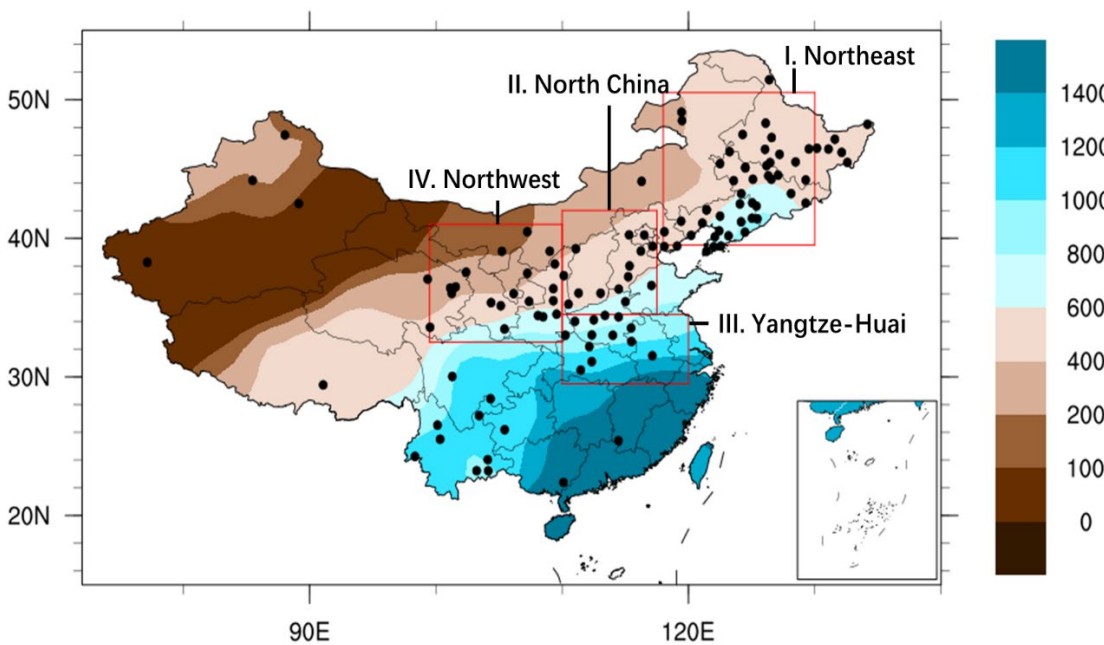


**Figure 1. Spatial distribution of 119 agricultural-meteorological observation stations and four research regions over China for the period 1981 to 2013. The colors denote the distribution of annual precipitation (unit: mm a$^{-1}$) from 1971-2000.**

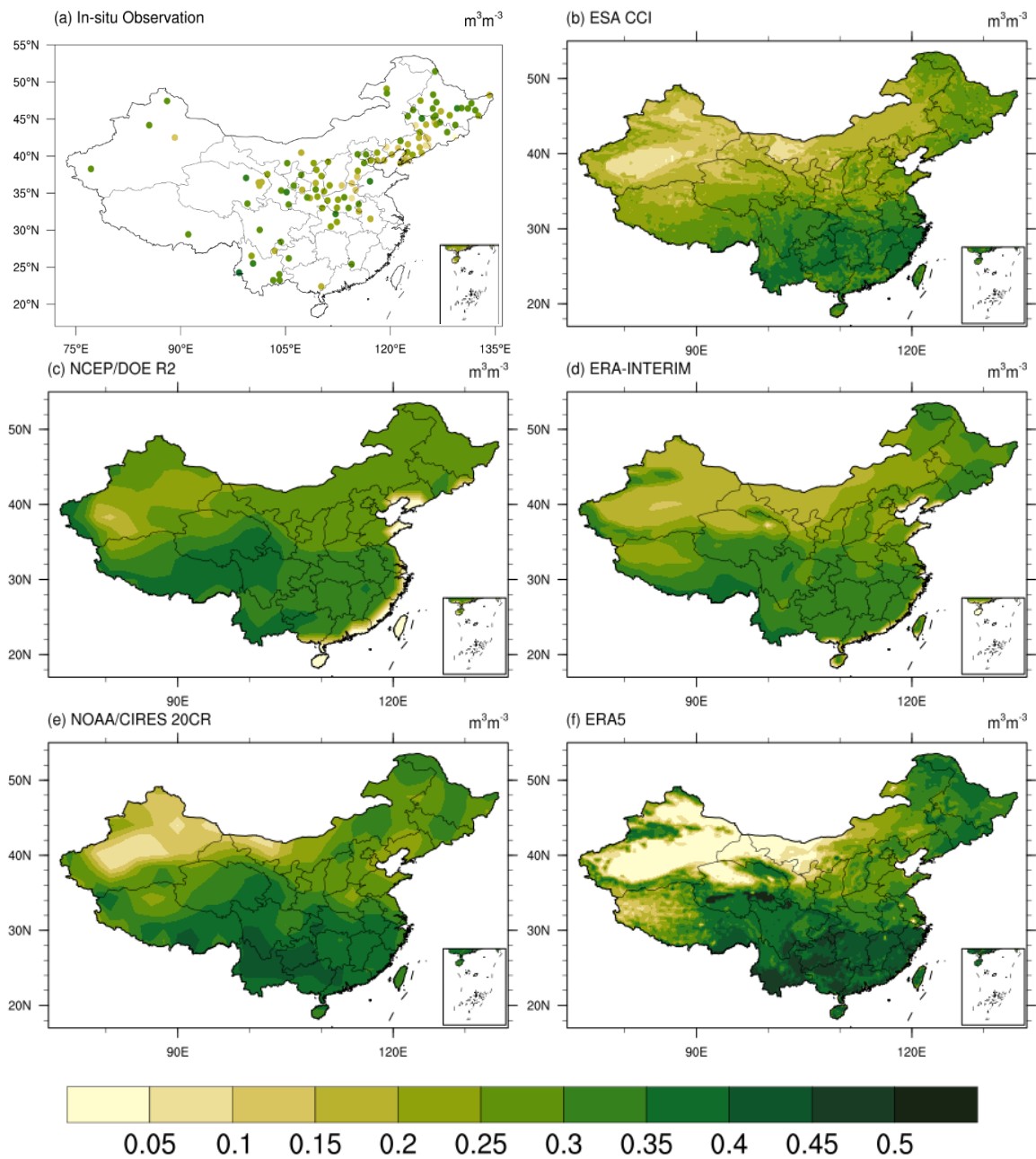

Figure 2. Annual averages of (a) observations and (b-e) five satellite/reanalysis SM products (units: m³·m⁻³) during June to August for the period of 1981 to 2013 in China.

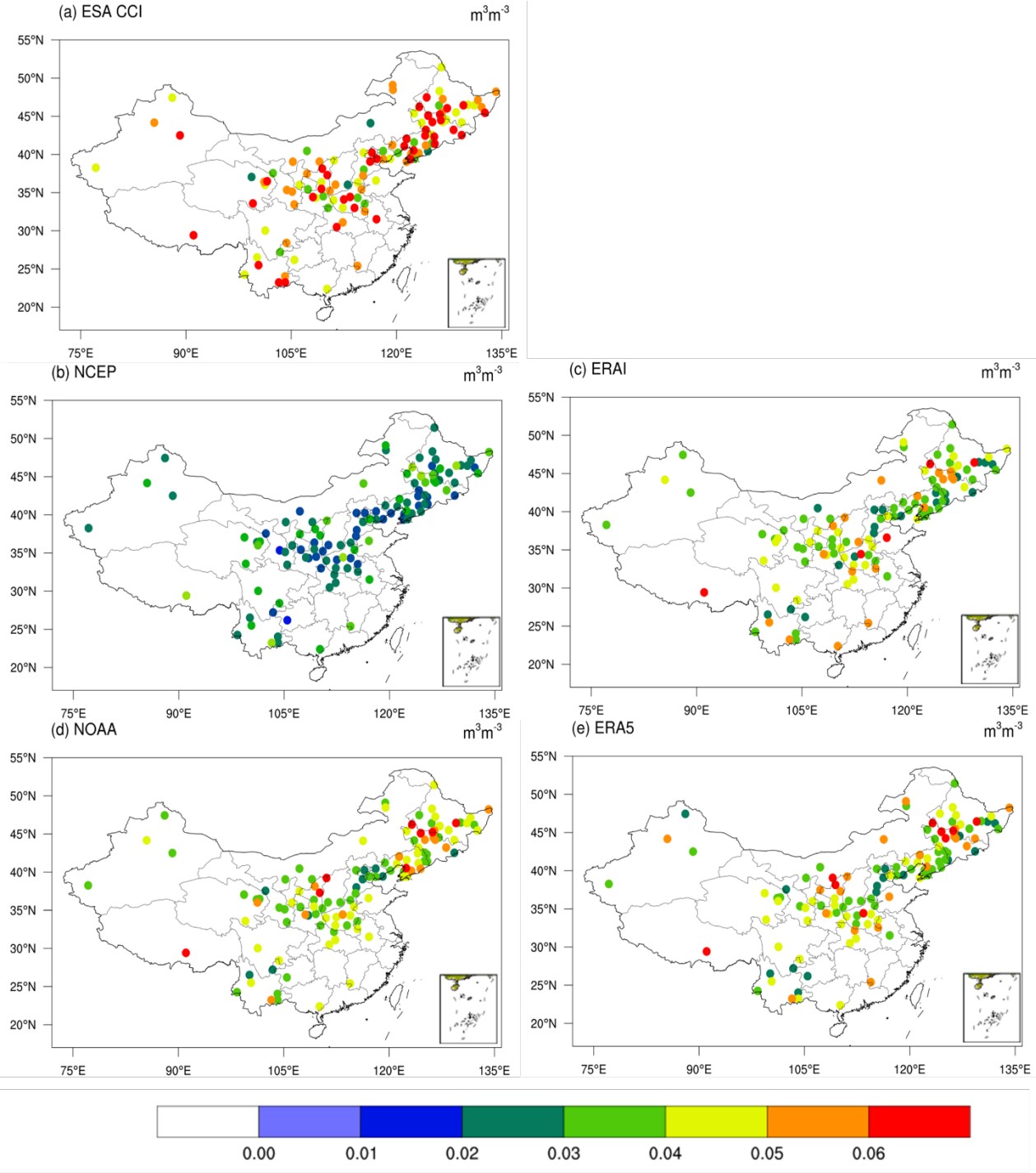

**Figure 3. Same as Figure 2 but for ubRMSE.**

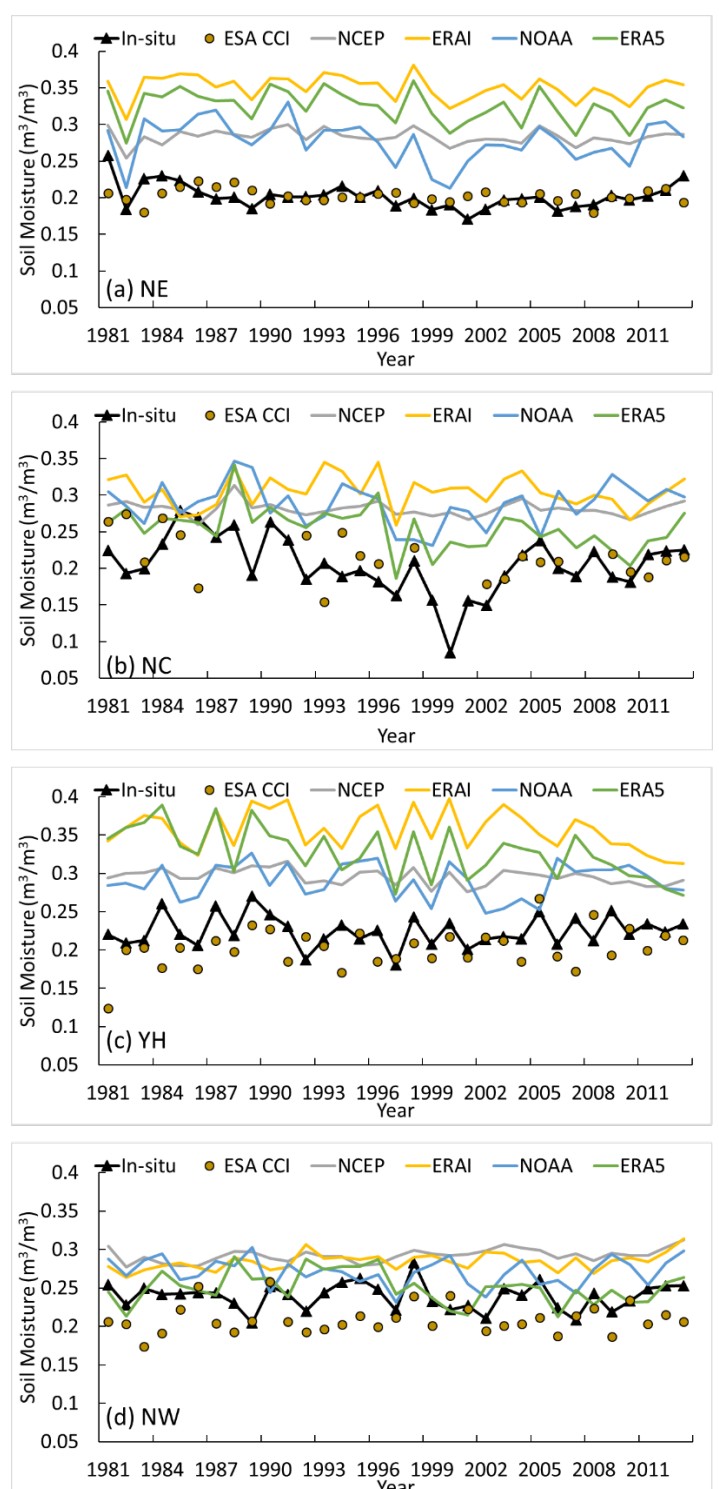

**Figure 4. Time series of SM in four research regions (a-d) from 1981 to 2013.**

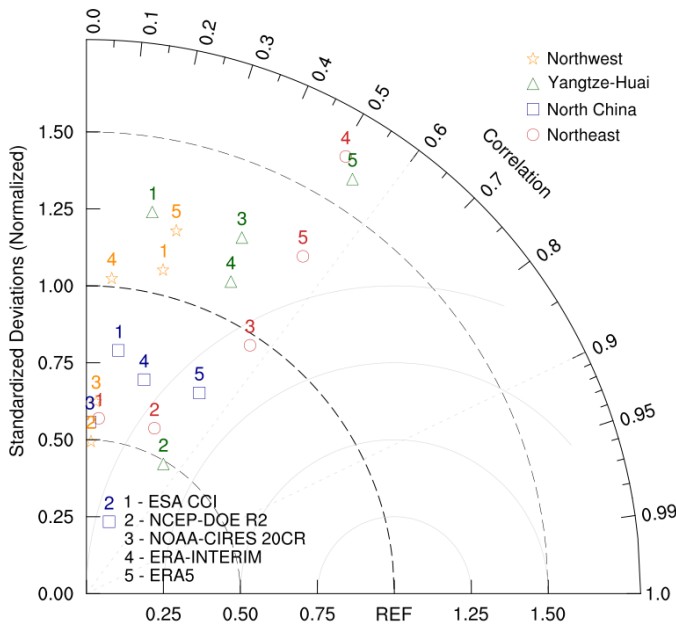

**Figure 5. Taylor diagrams of the comparison between multisource SM products and in situ observations. Ref. is the SM from in situ observations.**

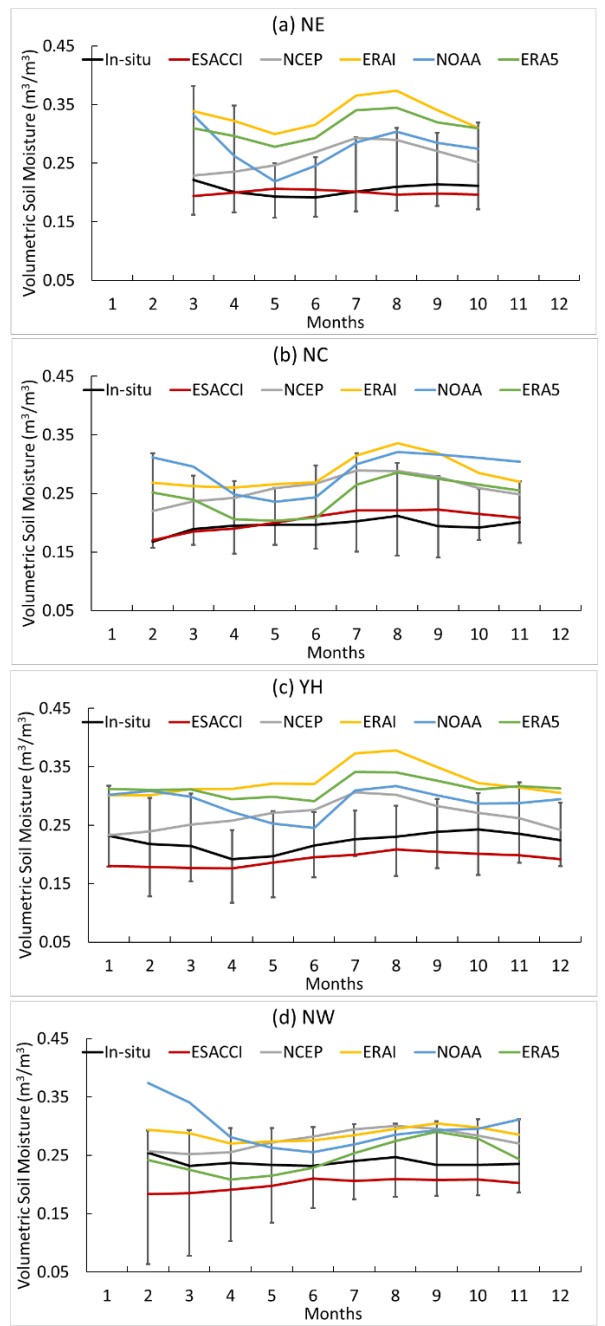

**Figure 6. Seasonality of SM distributions based on in situ observations and five products averaged over the (a) NE, (b) NC, (c) YH, and (d) NW regions from 1981 to 2013.**

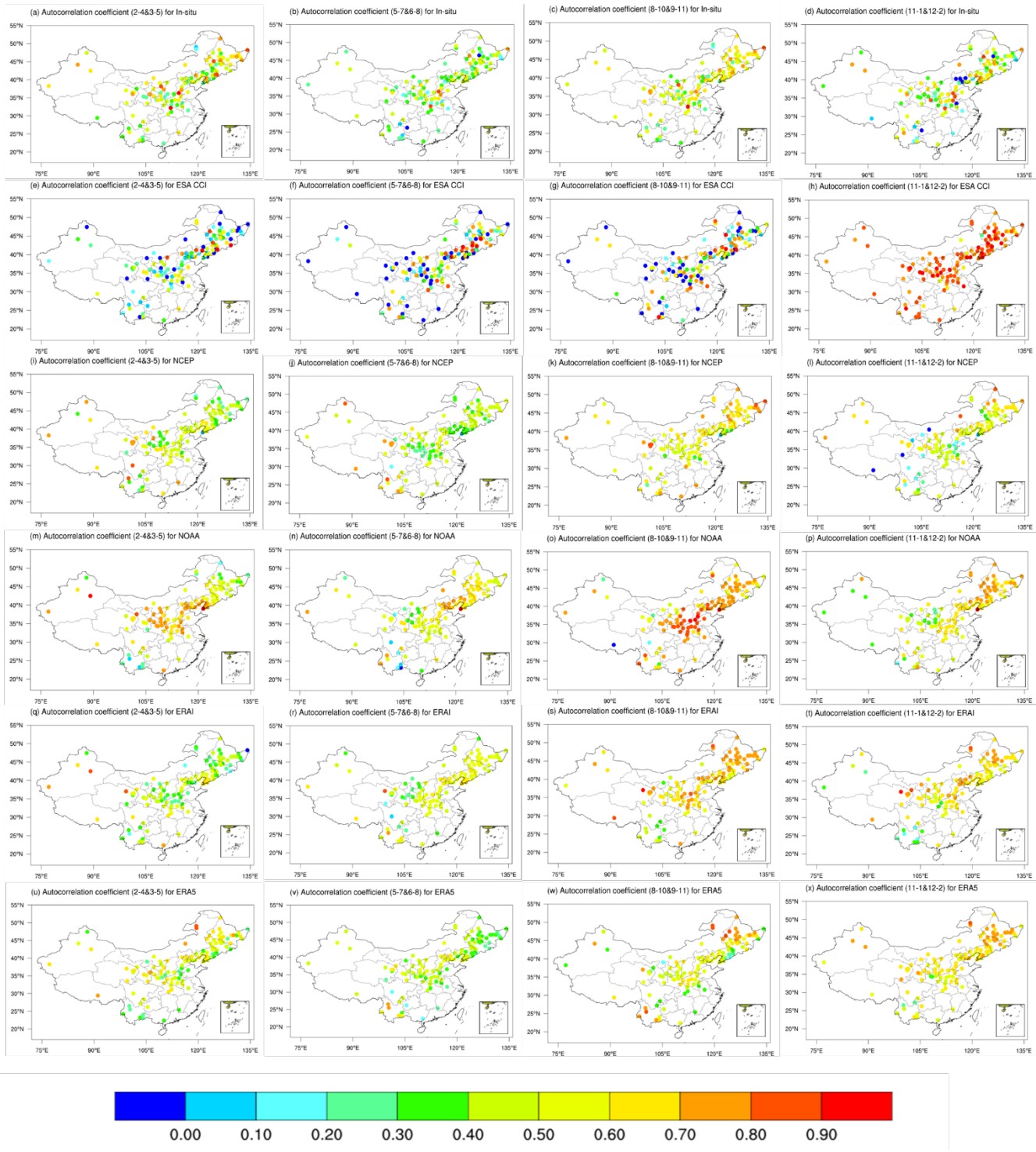

**Figure 7. Distribution of autocorrelation coefficient of SM in seasons: (1st row) FMA and MAM; (2nd row) MJJ and JJA; (3rd row) ASO and SON; (4th row) NDJ and DJF.**

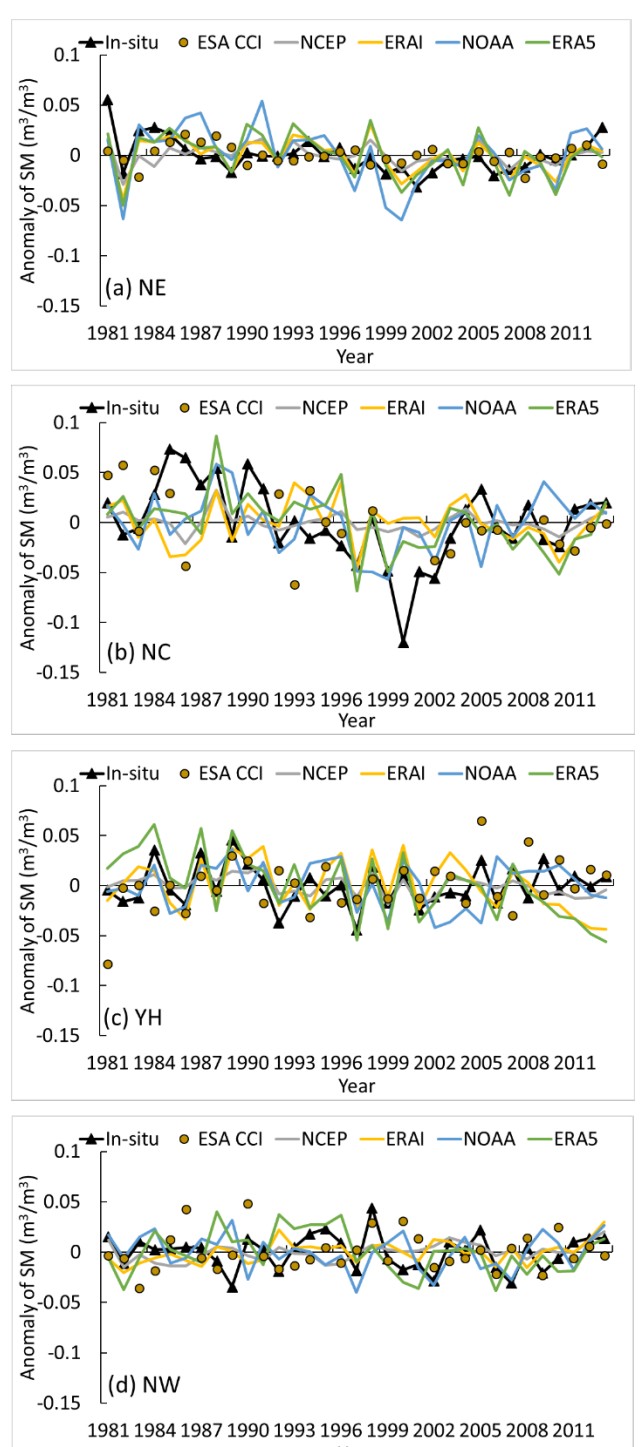


**Figure 8. Temporal evolution of the JJA SM anomaly time series from observations and the five satellite/reanalysis products in four research regions from 1981 to 2013.**

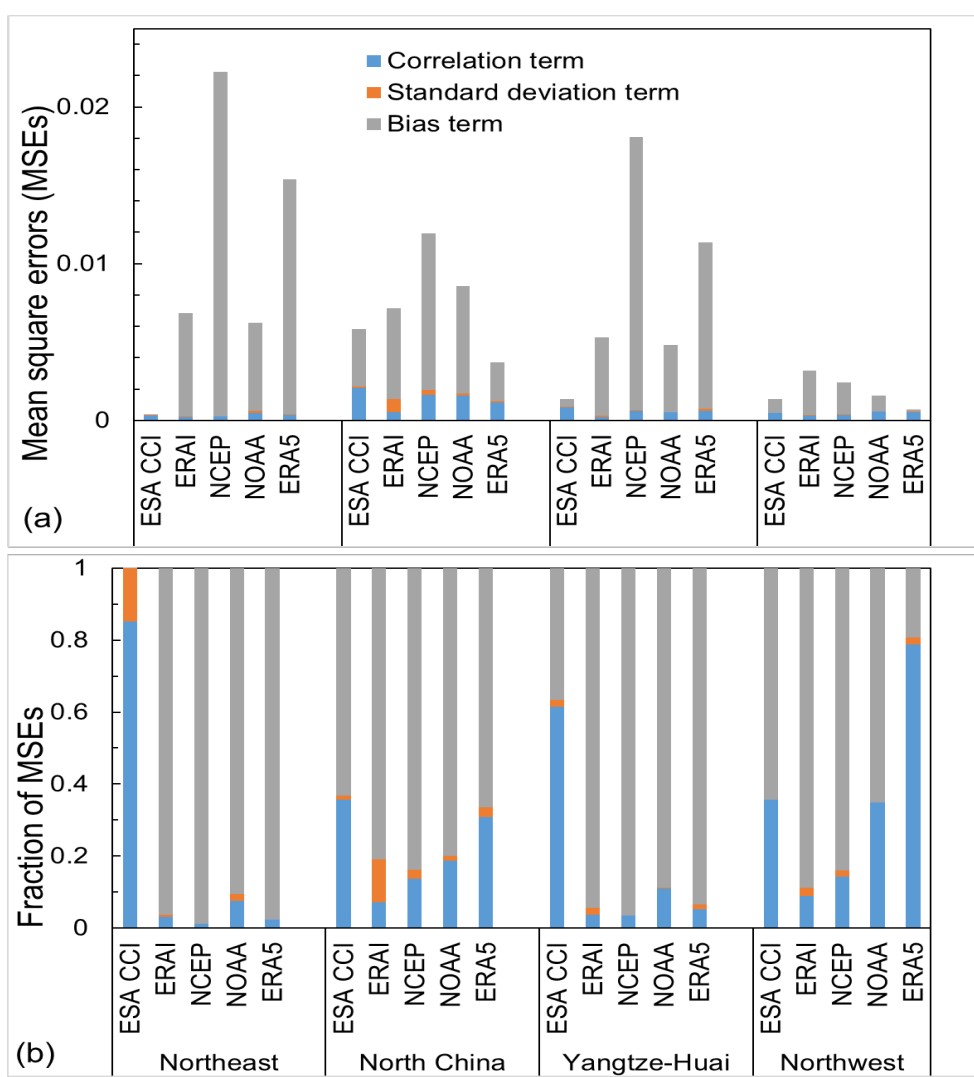

**Figure 9. The (a) decomposition of three terms to the mean square errors (MSEs) for the four satellite/reanalysis products from 1981 to 2013, as well as (b) their fraction.**

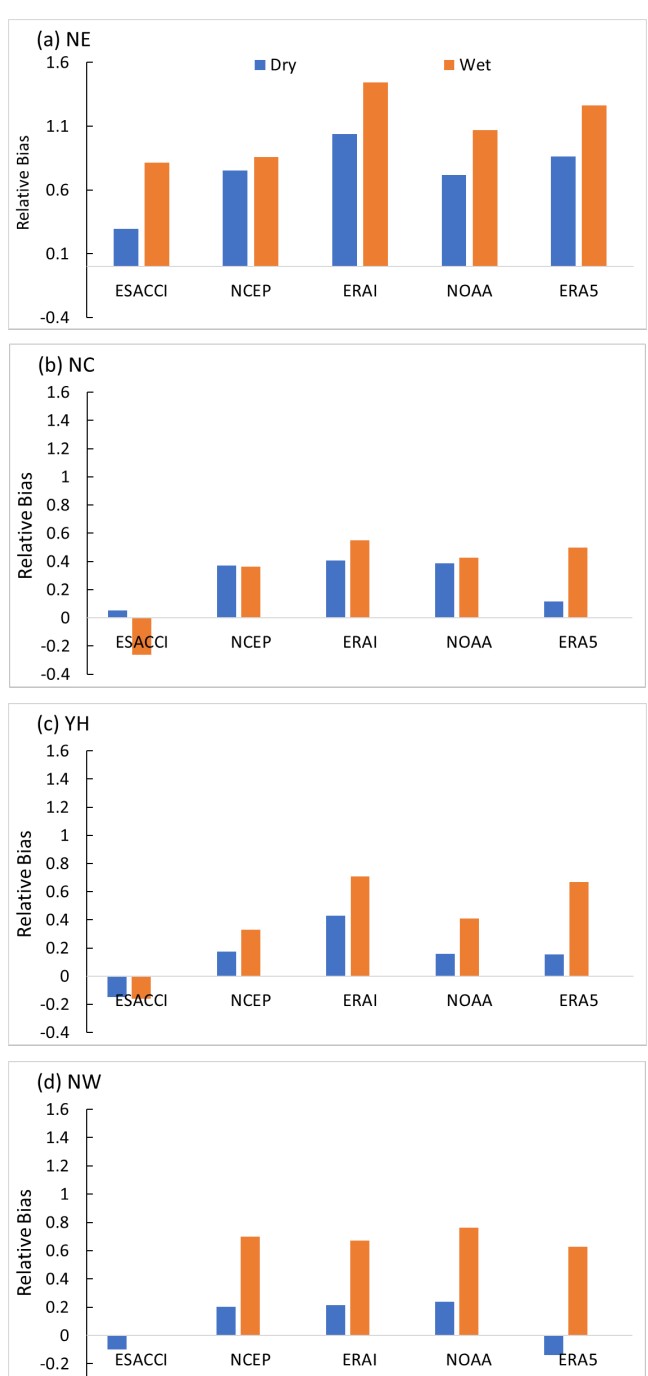

**Figure 10. The relative Bias of remote sensing and reanalysis SM against in situ observations under dry or wet conditions. The Dry condition consists of extreme (scPDSI<-4) and severe (scPDSI<-3) drought conditions, the wet condition consists of extreme (scPDSI>4) and severe (scPDSI>3) wet spell conditions.**


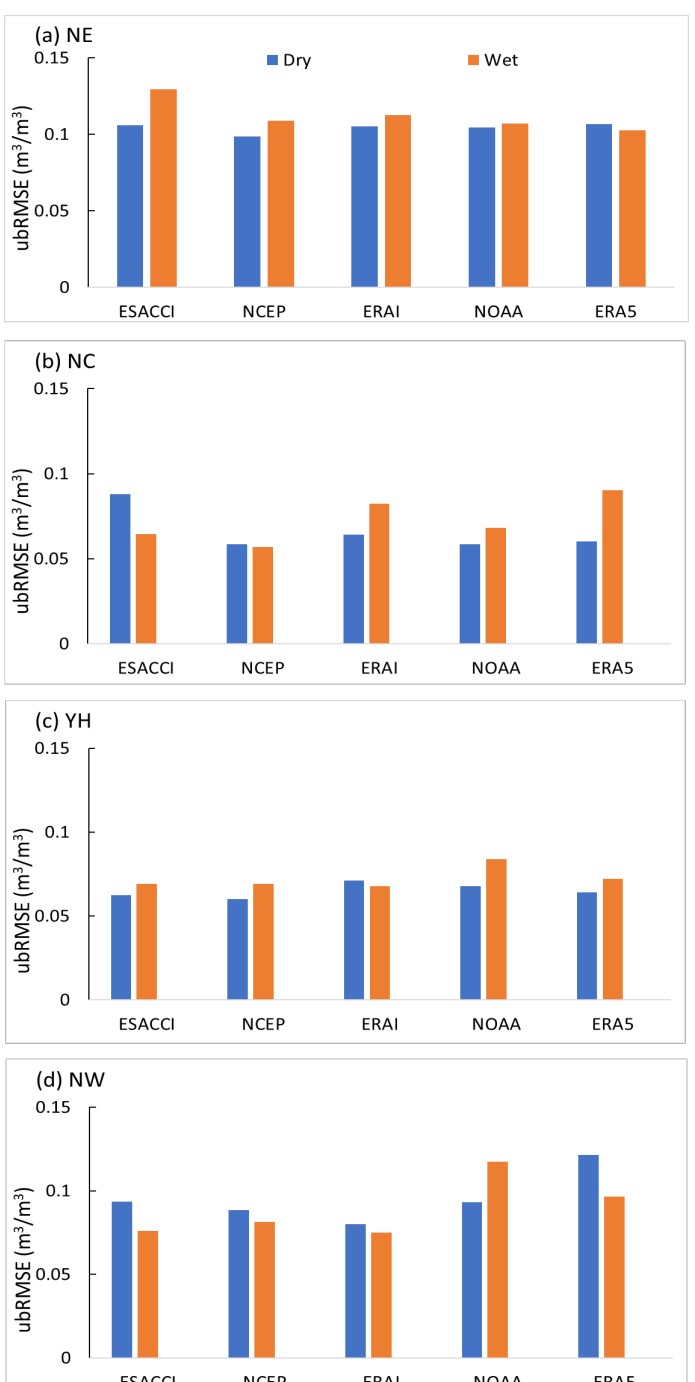

**Figure 11. The ubRMSE of remote sensing and reanalysis SM against in situ observations under dry or wet conditions in different regions. The Dry condition consists of extreme (scPDSI<-4) and severe (scPDSI<-3) drought conditions, the wet condition consists of extreme (scPDSI>4) and severe (scPDSI>3) wet spell conditions.**

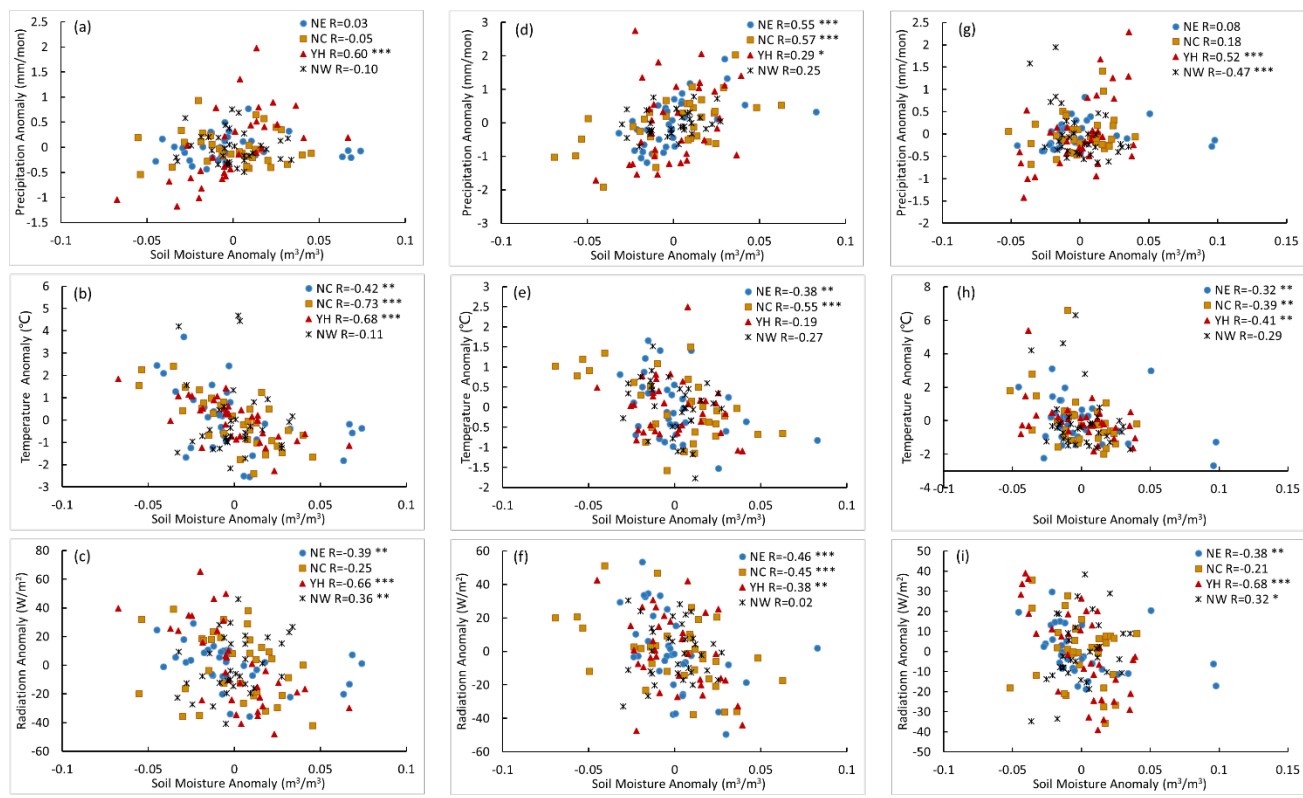

**Figure 12. Scatterplots of monthly anomalies of (a, d, g) precipitation, (b, e, h) temperature, and (c, f, i) net radiation vs observed soil moisture in the top 10 cm depth during 1981-2013 during (a, b, c) MAM, (d, e, f) JJA and (g, h, i) SON seasons. R is the correlation coefficient over four research regions, and the values marked with \*, \*\*, and \*\*\* means that the correlation coefficient has passed the significance test of 90 %, 95 % and 99 %, respectively.**


**Tables**

                              **Table 1: Details of the SM products used in the study.**

| Name | Soil Depths (cm) | Spatial Resolution | Temporal Resolution | Temporal Coverage |
|---|---|---|---|---|
| *In situ* | | | | |
| ISMN | 10, 20, 50, 70, 100 | | 3x monthly | 1981.01-1999.12 |
| Agricultural-meteorological stations | 10, 20, 50, 70, 100 | Total 778 stations (119 used) | 3x monthly | 1991.05-2013.12 |
| Mass percent of measured SM | 0-5, 5-10, 10-20, 20-30, 30-40, 40-50, 50-60, 60-70, 70-80, 80-90, and 90-100 | | 3x monthly | 1981.01-1998.12 |
| *Satellite* | | | | |
| ESA CCI | - 2-5 | 0.25°*0.25° | Daily, monthly | 1978.11-present |
| *Reanalysis* | | | | |
| ERAI | 0-7, 7-28, 28-100, 100-255 | 0.75°*0.75° | 4x daily, monthly | 1979.01-present |
| NCEP | 0-10, 10-200 | T62 (-2°*2°) | 4x daily, monthly | 1979.01-present |
| NOAA | 0, 10, 40, 100 | 2°*2° | 8x daily, monthly | 1851.01-2014.12 |
| ERA5 | 0-7, 7-28, 28-100, 100-255 | 0.28125°*0.28125° | 2x daily, monthly | 1979.01-present |

**Table 2: Names and spatial coverage of the selected research regions.**

|     |     | Regions       | Zonal Coverage (°E) | Meridional Coverage (°N) |
| --- | --- | ------------- | ------------------- | ------------------------ |
| I   | NE  | Northeast     | 118-130             | 39.5-50.5                |
| II  | NC  | North China   | 110-117.5           | 34.5-42                  |
| III | YH  | Yangtze-Huai  | 110-120             | 29.5-34.5                |
| IV  | NW  | Northwest     | 99.5-110            | 32.5-41                  |

**Table 3: Correlation coefficients, biases and RMSEs of the five datasets for JJA SM from 1981 to 2013. The coefficients in brackets are those that cannot pass the significance test (α=0.1) with n=33.**

| Regions | Products | Bias | RMSD | ubRMSE | Correlation |
|---------|----------|------|------|--------|-------------|
| Northeast | ESA CCI | 0.000 | 0.019 | 0.019 | (0.070) |
| | NCEP | 0.081 | 0.083 | 0.016 | 0.380 ** |
| | ERAI | 0.148 | 0.149 | 0.016 | 0.550 *** |
| | NOAA/CIRES 20CR | 0.075 | 0.079 | 0.024 | 0.509 *** |
| | ERA5 | 0.123 | 0.124 | 0.019 | 0.538 *** |
| North China | ESA CCI | -0.061 | 0.122 | 0.106 | (0.122) |
| | NCEP | 0.076 | 0.084 | 0.037 | (0.085) |
| | ERAI | 0.100 | 0.109 | 0.044 | (0.109) |
| | NOAA/CIRES 20CR | 0.083 | 0.093 | 0.041 | (0.093) |
| | ERA5 | 0.050 | 0.061 | 0.035 | (0.061) |
| Yangtze-Huai | ESA CCI | -0.022 | 0.037 | 0.029 | (0.173) |
| | NCEP | 0.071 | 0.073 | 0.017 | 0.510 *** |
| | ERAI | 0.132 | 0.134 | 0.025 | 0.398 ** |
| | NOAA/CIRES 20CR | 0.065 | 0.069 | 0.023 | 0.415 ** |
| | ERA5 | 0.103 | 0.107 | 0.027 | 0.535 *** |
| Northwest | ESA CCI | -0.030 | 0.037 | 0.022 | (0.227) |
| | NCEP | 0.053 | 0.056 | 0.019 | (0.027) |
| | ERAI | 0.045 | 0.049 | 0.020 | (0.048) |
| | NOAA/CIRES 20CR | 0.032 | 0.039 | 0.023 | (0.080) |
| | ERA5 | 0.011 | 0.026 | 0.023 | (0.244) |