# Peer review of "Comprehensive evaluation of satellite-based and reanalysis soil moisture products using in situ observations over China"

_Hydrology and Earth System Sciences, 2020_

## Author Comment (AC3)

**Community Comments**

1. The authors need to reappraise their motive of this study, because NOAA and NCEP soil moisture (SM) products (a spatial resolution of 2 degrees) are usually not qualified for hydro-meteorological studies (flood or drought as reviewed by Peng et al. 2020, in Remote Sensing of Environment) in mainland China. As pointed out by the other reviewer, such coarse spatial resolutions would cause representativeness errors. Although spatial averaging to some extent can alleviate such an effect, I still think errors of representativeness (together with differences in effective soil depth) might contribute substantially to the bias values. That is probably the reason why CCI (0.25 degrees) and ERA-5 (31 kilometers) have a slightly better performance.

**Response**: During this revision, the bias error caused by the mismatch of spatial representativeness between in situ data and all SM products has been removed by introducing the unbiased root mean square error (ubRMSE) (see Figure 3, Figure 11 and Table 3). Furthermore, the comparison was conducted at regional scales by calculating the reginal average of monthly value for all SM products, which can reduce the uncertainty caused by grid mismatch to some extent.

2. The presentation of results should be improved. In numerous cases, the authors repeat the overestimation of modelling SM data and the underestimation of remotely sensed SM data.

**Response:** Thanks for your suggestion, and we have refined the presentation of the results.

**Line 237-245:** *Generally, most SM products are able to capture the overall spatial distribution of the SM value, although the NOAA SM is highly overestimated all through the region. According to the in situ observations, SM is the lowest in the northwest and increases to the northeast and southeast. Except for NCEP, all the other datasets are able to represent the wet center in the northeast of China. SM is underestimated by ESA CCI, but overestimated for all the analysis datasets, except in Northwest China. In the ERA5 dataset, the region in the north of Northwest China is much drier than the other products, with average value less than 0.05 m³/m³. ERAI and ERA5 SM products are able to represent the decreasing trend from southeast to northwest, which is failed for the NCEP SM. The*

*largest biases reaching 0.15 m³/m³ are found in southern Northeast China, and the largest inconsistency is found in the northwest.*

**Line 257-262:** *Generally, all the reanalysis products have positive bias of 0.08~0.15 m3/m3, 0.05~0.10 m³/m³, 0.07~0.13 m³/m³, and 0.01~0.05 m³/m³ in the NE, NC, YH, and NW regions, respectively. ESA CCI tend to have negative bias with observations around -0.06~0 m³/m³. All products perform well in the NW region, and the worst performance is found in the NC region. ERAI largely overestimates SM in all the research regions, while NOAA and NCEP SM has the lowest bias among the reanalysis datasets. Reanalysis can better reproduce the variation characteristics than remote sensing during extreme events period, probably due to large percent of missing data, and instrument constrict.*

3.  Some descriptions contradict each other throughout the manuscript. For example, in Lines 188-190, the authors first report the underestimation in northwest China and then report the opposite side.

**Response:** There was a typo, which we have corrected as follows:

**Line 239-240:** *Except for NCEP, all the other datasets are able to represent the wet center in the northeast of China.*

4.  In Line 79, the authors promise to discuss on sources of SM errors. However, most of the explanations are speculations and even key words. In Line 223 for example, why different land surface types and varying soil parameters cause differences between CCI and model outputs? In Line 227, how vegetation presence leads to a clear SM seasonal cycle? In Line 237, how precipitation and frozen soils increase autocorrelation? Then in the following sentence, what particular soil type and texture decreases autocorrelation?

**Response**: Earlier studies have showed that low soil moisture content at top layer are associated with low precipitation and high evaporation (Jasper et al., 2006; Harmsen et al., 2009). Furthermore, the land surface vegetation and soil texture also play an important role. We add some details on the different land surface types and varying soil parameters as follows:

**Line 279-285:** *The difference in ESA CCI is smaller than all reanalysis products, especially in the period where in situ SM value is low, which is in line with Ma et al. (2019) that ESA*

*CCI have relative poor skills with lower time series correlations in sparse or dense VOD conditions but good performance in moderate densely vegetated areas (Zeng et al., 2015). Furthermore, soil types (silt, clay, sand) also plays an important role in terms of different regions. Chakravorty et al. (2016) studied the influence of soil texture on regional scale performance and found that large fractional RMSE is associated with large percentage of sand, might be one of the reasons that poor performance is found in the NW region.*

**Line 287-288:** *Seasonal cycle of SM in the NE region is obvious, partly due to the sufficient water content there.*

**Line 300-301:** *The lowest autocorrelation coefficient is found in the NW region, possibly because of the particular sand soil with relative high porosity and low water holding capacity.*

What is worth to say, our result is that the SM autocorrelation is low in summer and winter, indicating that the SM during these seasons are more easily influenced by precipitation and snow.

**Related references:**

Jasper K, Calanca P, Fuhrer J. Changes in summertime soil water patterns in complex terrain due to climatic change. Journal of Hydrology, 2006, 327: 550-563.

Harmsen E W, Norman L M, Nicole J S, J E Gonzalez. Seasonal climate change impacts on evapotranspiration, precipitation deficit and crop yield in Puerto Rico. Agricultural Water Management, 2009, 96: 1085-1095.

5.  Section 3.2.2, how autocorrelation is related to the performance of soil moisture products.

**Response**: The aim of this figure is to study the soil moisture memory in different seasons. **Line 306-308:** *The information of soil moisture autocorrelation gives hint for the assimilation of surface soil moisture into land surface models (Crow and Van den Berg, 2010), in which during summer and winter, some other related meteorological elements should be considered.*

**Related references:**

Crow, W., and Van den Berg, M.: An improved approach for estimating observation and model error parameters in soil moisture data assimilation, Water Resources Research, 46(12), doi:10.1029/2010WR00940, 2010.

6. Lines 288-289. Is this a manifestation of scaling effect? Spatial averaging (coarse resolution) masks out extremely low and high SM values.

**Response**: We have added the following supplementary information in the revised manuscript.

**Line 343-345:** *Figure 10 shows the rBias under different humid/arid conditions by utilizing SC-PDSI (Wells et al., 2004). The rBias of JJA SM between in situ observation and remote sensing/reanalysis was calculated at each in situ grid point as the bias divided by the mean of in situ observations, and then averaged over regions.*

7. Line 45, temporarily should be temporally.

**Response**: Corrected, thanks. **(Line 47)**

8. Line 65-66, this sentence makes no sense.

**Response**: This sentence was deleted.

9. Line 86, plus is incorrect here.

**Response**: "with a spatial resolution of 0.25° plus 0.25°" has been corrected into "with a spatial resolution of 0.25°". **(Line 85)**

10. Line 87, delete underlying.

**Response**: The word of "underlying" has been deleted as suggested. **(Line 88)**

11. The method section should provide more details, such as data interpolation in the vertical direction. The CCI has a penetration depth of < 2 cm, and the effective soil depth for model outputs is 0-10 cm, and the in-situ measurement depth is 10 cm. Such differences might also cause representativeness errors.

**Response**: The detailed information about the operation of measurements has been added in the revised manuscript, and also listed in Table 1.1.

**Line 132-135:** *The ISMN provides a global in-situ soil moisture database, which has been widely used for validation of satellite products and model simulation (e.g. Albergel et al., 2012). The SM data at the depth of 0~5 cm and 5~10 cm was obtained and averaged as the value at the depth of 0~10 cm.*

**Line 140-141:** *The SM data was observed at the depth of 10 cm, 20 cm, 50 cm, 70 cm, and 100 cm using drying methods, with the data at 10-cm depth utilized.*

**Line 153-155:** *The SM mass percent was measured at 11 levels with the depth of 0~5 cm, 5~10 cm, 10~20 cm, 20~30 cm, 30~40 cm, 40~50 cm, 50~60 cm, 60~70 cm, 70~80 cm, 80~90 cm, and 90~100 cm, in which the value at 10 cm depth are calculated as the average of the values at the depth of 5~10 cm and 10~20 cm.*

**Line 156-159:** *Considering that the field capacity and the dry bulk density are not measured at all stations, data from 119 stations are selected from 1981 to 2013. Not all in situ data were suitable for evaluation given instrumental error and observational conditions, for example, the available measurement period, installation depth and sensor placement. Therefore the evaluation was conducted in unfrozen and snow-free seasons, such as June-July-August (JJA).*

Furthermore, the representativeness errors have been talked about in the Discussion section.

**Line 368-373:** *ESA CCI SM product showed the top layer soil content at 5-cm depth or so. The in-situ measurement depth and model output are at the depth of 0-10cm, which were also treated as the top layer soil content. Such difference would also cause representativeness errors. Previous studies have found that there is a close relationship between surface SM and SM in the upper ten centimetres (i.e., Albergel et al., 2008; Dorigo et al., 2015), so the SM measurements at the depth of 10 cm were chosen as the reference to evaluate satellite-based and reanalysis products. Furthermore, introducing ubRMSE and conducting comparison at regional scale can remove the bias error caused by mismatch of grid cell to some extent.*

**Related references:**

Albergel, C., Rüdiger, C., Pellarin, T., Calvet, J. C., Fritz, N., Froissard, F., et al. (2008). From near-surface to root-zone soil moisture using an exponential filter: An

assessment of the method based on in-situ observations and model simulations. Hydrology and Earth System Sciences, 12, 1323–1337.

Dorigo, W. A., Gruber, A., De Jeu, R. A. M., Wagner, W., Stacke, T., Loew, A., Albergel, C., Brocca, L., Chung, D., Parinussa, R. M., and Kidd, R.: Evaluation of the ESA CCI soil moisture product using ground-based observations, Remote Sens. Environ., 162, 380–395, https://doi.org/10.1016/j.rse.2014.07.023, 2015.

12. Line 166, climate should be climatological.

**Response**: Corrected. **(Line 92)**

13. Line 178, Discussions should be Discussion.

**Response**: Corrected, thanks. **(Line 232)**

14. Lines 184-185, this sentence has been already in the previous section, and obviously does not belong to Result section.

**Response**: We didn't find this sentence in the previous section, and reserved this sentence. (**Line 234-236**)

15. Line 199, improper use of According to.

**Response**: This sentence has been changed as follows.

**Line 253-254:** *As referred in Table 2, all temporal variabilities of SM are averaged over the Northeast China, North China, Yangtze-Huai, and Northwest China regions, which are abbreviated as NE, NC, YH, and NW, respectively, below.*

16. Line 214, what kind of mechanism?

**Response**: The interpretation has been deleted.

17. Line 217, what is variability performance?

**Response**: This sentence has been changed into "implying a good performance of variability". **(Line 273)**

18. Lines 217-18, this sentence "demonstrating…" makes no sense.

**Response**: This sentence has been changed into "demonstrating that both products represent poor performance of changing characteristics." (**Line 273-274**)

19. Line 232, the snow-covered and frozen grids were not removed in this study?

**Response**: In the former manuscript, we only discarded in situ soil moisture data during snow or frozen days. During this revision, the months with large percent of frozen and snow days were discarded for comparison. Furthermore, if the in situ observation were missing, all reanalysis data at the same period were also treated as missing value.

**Line 157-159:** *Not all in situ data were suitable for evaluation given instrumental error and observational conditions, for example, the available measurement period, installation depth and sensor placement. Therefore the evaluation was conducted in unfrozen and snow-free seasons, such as June-July-August (JJA).*

**Line 227-231:** *The comparisons were performed as follows: (i) make a correspondence between all soil moisture data sets and in situ SM by using the values at the nearest neighbor grids; (ii) compare all the SM products at regional scales by calculating the reginal average of monthly value of all SM products, which has been proved can reduce the uncertainty caused by grid mismatch to some extent (Nie et al., 2008); (iii) if the in situ observation were missing, all reanalysis data at the same period were also treated as missing value, which were not taking into account.*

**Related references:**

Nie, S., Luo, Y., Zhu, J.: Trends and scales of observed soil moisture variation in China, Advance in Atmosphere Science, 25, 43–58, 2008.

20. Line 300, the explanations are unclear and confusing.

**Response:** The explanation was improved by integrating the relationship between net radiation and evaporation.

**Line 358-366:** *Previous studies have showed that soil moisture is influenced by the combination of precipitation and evaporation, in which land surface evaporation is linked with temperature and surface net radiation (Jasper et al., 2006; Harmsen et al., 2009). Figure 12 shows scatter plots of (a, d, g) precipitation, (b, e, h) temperature, and (c, f, i)*

*net radiation anomalies versus observed SM anomalies over different regions in (left column) MAM, (middle column) JJA, and (right column) SON seasons. Obvious positive correlations are found between precipitation and SM in the YH regions during MAM and SON seasons, and in the NE and NC regions during JJA season. Temperature and net radiation show negative correlation with in the NE, NC, and YH regions. The correlation coefficient is low for all meteorological variables in the NW region, which may be attributed to the special soil type there. Soil moisture in the NE and NC regions tends to be influenced by temperature during cold seasons. SM in the YH region tend to be influenced by radiation during warm seasons, due to the large evaporation there.*

**Related references:**

Jasper K, Calanca P, Fuhrer J. Changes in summertime soil water patterns in complex terrain due to climatic change. Journal of Hydrology, 2006, 327: 550-563.

Harmsen E W, Norman L M, Nicole J S, J E Gonzalez. Seasonal climate change impacts on evapotranspiration, precipitation deficit and crop yield in Puerto Rico. Agricultural Water Management, 2009, 96: 1085-1095.

21. Line 321, it is not quite right to say "CCI is not useful".

**Response:** This sentence has been changed as follows:

**Line 409-410:** *However, ESA CCI shows poor performance in terms of its low correlation and missing values, especially in Northeast China.*

22. Why not use GLDAS (the same grid resolution as CCI) or CLDAS (more spatial details) data as validation reference? Although with a shorter temporal coverage, other optimized SM data in mainland China can also serve as references. These data reduce representativeness errors.

**Response:** Thanks for your advice. Firstly, this study is focus on the long-term evaluation, so those products with shorter temporal coverage were not considered in this study. Secondly, the estimation using GLDAS and CLDAS data as reference will be considered in the further study.

---

## Author Response (AR1)

**Response to Editor:**

**Dear Editor,**

Thank you very much for your kind consideration and the constructive suggestion for improving this paper from the referees and community. We are submitting the revised manuscript entitled "Comprehensive evaluation of satellite-based and reanalysis soil moisture products using in situ observations over China" (HESS-2020-611).

During this revision, we have added four co-authors including Bo Qiu, Jun Ge, Kai Qin, and Yong Xue, who have contributed to the interpretation of the results and revision of the manuscript. According to the reviewers' comments, we have added more details about data uncertainty and representative, as well as the interpretation of the results. Enclosed please find the responses to the referees. We sincerely hope this manuscript will be finally acceptable to be published on Hydrology and Earth System Sciences.

We greatly appreciate your efforts in the review process of this paper. If you have any question about this paper, please don't hesitate to let me know. Best regards,

Sincerely yours,

Xiaolu Ling

**Referee Comments#1:**

**General comments:**

Soil moisture, as one of the essential climate variables, has attracted more and more attention from climate research. However, there is still a long way to go for the recently widely used soil moisture products, including reanalyses based on models and retrievals from remotely sensed data, to be comparable with observations. To further develop and properly use them, it is necessary to compare with in situ observations to reveal their uncertainties. In this manuscript, the five satellite-based and reanalysis soil moisture products were evaluated in China with in situ observations for top soil layer (0-10 cm). By now the manuscript still needs to further discuss the uncertainties of in situ observations of soil moisture data, the influence of sparse data samples, and thus the unfair to compare grid products using point-scale measurements. In particular, the author pointed out that the bias term controlled the deviations of soil moisture products from the observed values. This partly stems from the spatial mismatches in the comparisons of the soil moisture measured at a point with model grid means. So, it requires more discussion about its implications. In addition, the method part needs to provide more details, for example, how the monthly means were estimated using 3sample observations per month.

**Response:** We appreciate your comments, which are helpful for us to further improve this paper. In the revised manuscript, we have focused on the following issues.

- (1) More detailed information has been added in the revised manuscript, such as the combination of active and passive product (see response#2), improvement of ERA5 to ERAI (see response#3), how monthly means were estimated using 3-sample observations per month (see response#4), and so on.
- (2) In order to remove the bias error caused by the mismatch of spatial representativeness between in situ data and all SM products, the unbiased root mean square error (ubRMSE) was introduced to evaluate temporal dynamic variability (see Figure 3(Line 730), Figure 11 (Line 760) and Table 3 (Line 775)). Furthermore, the comparison was conducted at regional scales by calculating the reginal average of monthly value for all SM products, which can reduce the uncertainty caused by grid mismatch to some extent.

(3) A more in-depth discussion especially focusing on the physical explanations has been added in the revised manuscript.

At the discussion section, uncertainties caused by comparing in situ observations with all products at different layers and grid mismatch are discussed.

Line 369-374: ESA CCI SM product showed the top layer soil content at 5-cm depth or so. The in-situ measurement depth and model output are at the depth of 0-10cm, which were also treated as the top layer soil content. Such difference would also cause representativeness errors. Previous studies have found that there is a close relationship between surface SM and SM in the upper ten centimetres (i.e., Albergel et al., 2008; Dorigo et al., 2015), so the SM measurements at the depth of 10 cm were chosen as the reference to evaluate satellite-based and reanalysis products. Furthermore, introducing ubRMSE and conducting comparison at regional scale can remove the bias error caused by mismatch of grid cell to some extent.

We further discuss why ESA CCI showed lower correlation with in situ observations.

Line 375-381: The ESA CCI combined data generally increase the number of observations available for a time period but the correlation coefficients were not better than those of the best performing single dataset (Dorigo et al., 2015). Dorigo et al. also studied the possible reasons of input data, and found that the low correlation of combined product possibly due to the merging procedure, including the influence of vegetation (Taylor et al., 2012), the different original overpass time, and the scaling of high resolution ASCAT product to lower resolution reference products. Beck et al. (2021) found that ESA CCI SM performed better in eastern Europe in terms of high-frequency fluctuations, and speculated the overall performance of ESA CCI may be not so good due to incorporating ASCAT that performed less well.

The physical explanations of spatio-temporal SM variation have also been added.

Line 389-394: Precipitation and evaporation are found to be the most important determinant of soil moisture simulation performance, in which the evaporation is associated with temperature and radiation (Gottschalck et al., 2005; Mall et al., 2006; Chen & Yuan, 2020). SM value in the analysis is overestimated, partly due to the reason that the JJA precipitation over China is overestimated by models (e.g., Luo et al., 2013; Yun et al., 2020). The largest bias of precipitation overestimation using the hourly 31-

km-resolution ERA5 reanalysis data is found over the Tibetan and Yun-Gui Plateaus, the North China Plain, and the southern mountains, which gives one the explanation why reanalysis products represent the worst performance over the NC region.

The detail about soil type and texture has been added.

Line 395-403: Soil type and soil texture are also important elements for soil moisture estimation. In the southwest of the NE region, the sand fraction of the topsoil can reach about 80%-90%, and the sand fraction and clay fraction of the topsoil are around 30%-40% and 10%-30% respectively (Shangguan et al., 2012) in the north NE region. The inconsistent of the soil types over the NE region might interpret why the large inconsistency of spatial distribution were found. In the northwest of the NW region, sand fraction is larger than 80%, and the sand fraction is low in the southeast of the NW region. The large difference of soil types over the north NW region is one of the reasons that all products show poor performance. In the NC and YH regions, sand and clay fraction of the topsoil account for about 10%-20% and 30%-50%, 30%-50% and 0-20% respectively. The different performance over the NC and YH regions gives hints that reanalysis products tend to performance worse when the soil contains large percentage of sand because of its high porosity and low water holding capacity.

**Specific comments and suggestions:**

 "mainland China" is NOT a right term, you can use: the Chinese Mainland, Mainland of China or China's Mainland.

**Response**: Thanks for your suggestion, and all the "mainland China" have been changed into "the Chinese Mainland" as suggested.

2. 2.1.1 ESA CCI SM, how the various retrievals of the passive and active sensors combined should be detailed a bit more, for example, using land surface model products?

**Response**: Thanks for your advice, and we have added more detail information about how the various retrievals of the passive and active sensors combined in the revised manuscript, as showed in Section 2.1.1. we also added the related reference.

**Line 84-96:** The ESA CCI SM v04.4 combined product is employed in this study, which provides SM data starting from November 1978 until June 2018 with a spatial

resolution of 0.25°. The project of ESA CCI is to use C-band microwave scatterometers (Aqua satellite and the Advance Scatterometer, ASCAT) and multi-channel microwave radiometers (SMMR, SSM/I, TMI, AMSR-E, WindSat, AMSR2) to produce a long-term reliable time series of SM (Chakravorty et al., 2016). The ESA CCI SM v4 is better at detecting SM changes (Balenzano et al., 2011) than previous versions, as it merges all active and passive Level 2 products directly to generate the combined product, rather than creating active and passive products separately and then merging together (ESA, 2018; Gruber et al., 2019). Global Land Data Assimilation System Noah (GLDAS 2.1) was used as a scaling reference in the combined product to obtain a consistent climatology, flagging of high vegetation optical depth (VOD) for Soil Moisture and Ocean Salinity (ESA SMOS) and AMSR-2 method changed (Dorigo et al., 2017; Pasik et al., 2020). A polynomial SNR-VOD regression and the p-value based mask was used to fill spatial gaps in TC-based SNR estimates, and exclude unreliable input dataset in the combined product, respectively. Here, we evaluate all the products over the period from 1981 to 2013 (the same as below), during which in situ measurements are also available. The top layer of ESA CCI SM data at the depth of  $2\sim5$  cm depth are estimated.

**Related references:**

- Balenzano, A., Mattia, F., Satalino, G., and Davidson, M. W. J.: Dense temporal series of C- and L-band SAR data for soil moisture retrieval over agricultural crops, IEEE J. Sel. Top. Appl. Earth Obs. Remote Sens., 4, 439–450, https://doi.org/10.1109/jstars.2010.2052916, 2011.
- Chakravorty, A., Chahar, B. R., Sharma, O. P., and Dhanya, C. T.: A regional scale performance evaluation of SMOS and ESA-CCI soil moisture products over India with simulated soil moisture from MERRA-Land, Remote Sens. Environ., 186, 514–527, https://doi.org/10.1016/j.rse.2016.09.011, 2016
- Dorigo, W., Wagner, W., Albergel, C., Albrecht, F., Balsamo, G., Brocca, L., Chung, D., Ertl, M., Forkel, M., Gruber, A., Haas, E., Hamer, P. D., Hirschi, M., Ikonen, J., de Jeu, R., Kidd, R., Lahoz, W., Liu, Y. Y., Miralles, D., Mistelbauer, T., Nicolai-Shaw, N., Parinussa, R., Pratola, C., Reimer, C., van der Schalie, R., Seneviratne, S. I., Smolander, T., and Lecomte, P.: ESA CCI soil moisture for improved earth system understanding: state-of-the art and future directions, Remote Sens. Environ., 203, 185–215, https://doi.org/10.1016/j.rse.2017.07.001, 2017.

- Gruber, A., Scanlon, T., van der Schalie, R., Wagner, W., and Dorigo, W.: Evolution of the ESA CCI Soil Moisture climate data records and their underlying merging methodology, Earth Syst. Sci. Data, 11, 717–739, https://doi.org/10.5194/essd-11-717-2019, 2019.
- Pasik, A., Scanlon, T., Dorigo, W., de Jeu, R.A.M, Hahn, S., van der Schalie, R., Wagner, W., Kidd, R., Gruber, A., Moesinger, L., Preimesberger, W.: ESA Climate Change Initiative Plus Soil Moisture: Algorithm Theoretical Baseline Document (ATBD) Supporting Product Version v05.2, Earth Observation Data Centre for Water Resources Monitoring (EODC) GmbH, TU Wien, VanderSat, CESBIO and ETH Zürich, pp: 71, 2020.

3. 2.1.5 ERA5 SM, the improvements of land processes in ERA5 against ERAI are helpful to understanding of the results with respect to in situ soil moisture in these two reanalysis.

**Response**: The main improvements of ERA5 against ERAI are listed as follows: (1) uses a new version of the ECMWF assimilation system IFS (IFS Cycle 41R2); (2) combines vast amounts of historical observations, including ozone, aircraft and surface pressure data, as well as various newly reprocessed datasets and recent instruments that could not be ingested in ERA-Interim; (3) includes more model input, for example, the World Climate Research Programme (WCRP) Coupled Model Intercomparison Project (CMIP) greenhouse gases, volcanic eruptions, sea surface temperature (SST), and sea-ice cover, which are appropriate for climate; and (4) has much higher spatial and temporal resolution. And the above information was added in Section 2.1.5.

Line 117-124: ERA5 is the latest reanalysis product produced by ECMWF, covering the period from 1979 to present. The product uses a new version of the ECMWF assimilation system IFS (IFS Cycle 41R2), and combines vast amounts of historical observations, including ozone, aircraft and surface pressure data, as well as various newly reprocessed datasets and recent instruments that could not be ingested in ERA-Interim (C3S, 2017). The ERA5 model input includes the World Climate Research Programme (WCRP) Coupled Model Intercomparison Project (CMIP) greenhouse gases, volcanic eruptions, sea surface temperature (SST), and sea-ice cover, which are appropriate for climate. Furthermore, the spatial (31 km globally) and temporal (hourly) resolutions of ERA5 are rather high compared to ERAI. ERA5 will eventually cover the period from 1950 to the present, and one of its key improvements is better SM (Komma et al., 2008).

4. 2.2 In Situ SM and Preprocessing of Datasets, the in situ observations were took from three datasets, so details about difference in the operation of measurements and the means of quality control for the datasets are necessary to assess the credibility of in situ data.

**Response**: The detailed information about the operation of measurements has been added in the revised manuscript, and also listed in Table 1.1.

**Line 132-135:** The ISMN provides a global in-situ soil moisture database, which has been widely used for validation of satellite products and model simulation (e.g. Albergel et al., 2012). The SM data at the depth of 0~5 cm and 5~10 cm was obtained and averaged as the value at the depth of 0~10 cm.

Line 140-141: The SM data was observed at the depth of 10 cm, 20 cm, 50 cm, 70 cm, and 100 cm using drying methods, with the data at 10-cm depth utilized.

Line 153-155: The SM mass percent was measured at 11 levels with the depth of  $0\sim5$  cm,  $5\sim10$  cm,  $10\sim20$  cm,  $20\sim30$  cm,  $30\sim40$  cm,  $40\sim50$  cm,  $50\sim60$  cm,  $60\sim70$  cm,  $70\sim80$  cm,  $80\sim90$  cm, and  $90\sim100$  cm, in which the value at 10 cm depth are calculated as the average of the values at the depth of  $5\sim10$  cm and  $10\sim20$  cm.

Line 156-159: Considering that the field capacity and the dry bulk density are not measured at all stations, data from 119 stations are selected from 1981 to 2013. Not all in situ data were suitable for evaluation given instrumental error and observational conditions, for example, the available measurement period, installation depth and sensor placement. Therefore the evaluation was conducted in unfrozen and snow-free seasons, such as June-July-August (JJA).

5. The 'CN05.1' should be defined before its first citation.

**Response**: The 'CN05.1' was defined as 'the station observational meteorology dataset (CN05.1)' in the revised manuscript (**Line 168-169**).

6. Line 155, 'different drought/well conditions', 'well' is a typing error?

**Response**: Sorry for the typo, and the "well conditions" has been changed into "wet conditions" (Line 173).

7. More detailed information on the decomposition of MSEs and the test methods is necessary for potential readers.

**Response**: The evaluated metrics (including bias, relative bias, the Pearson correlation coefficient, root mean square difference (RMSD), and the unbiased relative root mean square error (ubRMSE) has been added in Section 2.4.1 (see Equation (3) to (7)).

Line 180-194: The comparisons were conducted through the statistical metrics, such as the Bias, relative Bias (rBias), Pearson correlation coefficient (R), root mean square difference (RMSD), and the unbiased root mean square error (ubRMSE) using the following formulas:

$$Bias = \frac{\sum_{t=1}^{n} (x_{p,t} - x_{obs,t})}{n}$$
(3)

$$rBias = \frac{Bias}{Mean(Observation)}$$
(4)

$$R = \frac{\sum_{t=1}^{n} (x_{obs,t} - \mu_{obs})(x_{p,t} - \mu_p)}{\sqrt{\sum_{t=1}^{n} (x_{obs,t} - \mu_{obs})^2} \sqrt{\sum_{t=1}^{n} (x_{p,t} - \mu_p)^2}}$$
(5)

$$RMSD = \sqrt{\frac{\sum_{t=1}^{n} (x_{p,t} - x_{obs,t})^2}{n}}$$
(6)

$$ubRMSE = \sqrt{RMSD^2 - Bias^2} \tag{7}$$

in which n is the total number of time steps,  $x_{p,t}$  and  $x_{obs,t}$  is the value of SM products (including remote sensing and reanalysis) and observation at time-step t,  $\mu_{obs}$ and  $\mu_p$  are the mean of the in situ observed values and all SM products, Mean(observation) is the average of observation. The metrics of rBias was used to study the performance of various regions under different drought or wet conditions. The ubRMSE is introduced to evaluate temporal dynamic variability to get rid of the bias error caused by the mismatch of spatial representativeness between the in situ data and all SM products (Jackson et al., 2010, 2012; Entekhabi et al., 2014). What is worthy to say, the in situ observation were not considered as 'true' value because of instrumental errors and representativeness, so the RMSD terminology was used in this study.

**Related references:**

Entekhabi, D., et al. (2014), SMAP Handbook Soil Moisture Active Passive, Mapping Soil Moisture Freeze/Thaw From Space, 180 pp., Nat. Aeronaut. Space Admin., Jet Propul. Lab., Pasadena, Calif.

- Jackson, T., M. Cosh, R. Bindlish, P. Starks, D. Bosch, M. Seyfried, D. Goodrich, S. Moran, and J. Du: Validation of Advanced Microwave Scanning Radiometer soil moisture products, IEEE Trans. Geosci. Remote Sens., 48(12), 4256–4272, doi:10.1109/TGRS.2010.2051035, 2010.
- Jackson, T. J., et al.: Validation of Soil Moisture and Ocean Salinity (SMOS) soil moisture over watershed networks in the U.S., IEEE Trans. Geosci. Remote Sens., 50(5), 1530–1543, doi:10.1109/TGRS.2011.2168533, 2012.

The detailed information on the decomposition of MSEs was also added in Section 2.4.2 (see Equation (8) to (12)).

**Line 196-213:** To better explain the disagreement between all the SM products and in situ observations, the mean square errors (MSEs, as defined in Eq.(8)) of each product in individual regions are utilized. To decompose the MSEs, the Nash-Sutcliffe efficiency (NSE, Nash and Sutcliffe, 1970) are utilized as defined in Eq.(9).

$$MSE = \frac{1}{n} \sum_{t=1}^{n} (x_{p,t} - x_{obs,t})^2$$
(8)

NSE =
$$1 - \frac{\sum_{t=1}^{n} (x_{p,t} - x_{obs,t})^2}{\sum_{t=1}^{n} (x_{obs,t} - \mu_{obs})^2} = 1 - \frac{MSE}{\sigma_{obs}^2}$$
 (9)

NSE was decomposed as the correlation, the conditional bias, and the unconditional bias as showed in Eq.(9) (Murphy, 1988).

$$NSE = A - B - C \tag{10}$$

 $A = R^2$

$$\mathbf{B} = [\mathbf{R} - (\sigma_p / \sigma_{obs})]^2$$

$$C = [(\mu_p - \mu_{obs})/\sigma_{obs}]^2$$

in which R is the correlation coefficient of observations and products,  $\sigma_{obs}$  and  $\sigma_p$  are the standard deviation of in situ data and all SM products. The Eq.(10) can be transformed as Eq.(11), representing the correlation, the bias and the variability.

$$NSE = 2 \cdot \alpha \cdot R - \alpha^2 - \beta_n^2 \tag{11}$$

$$\alpha = \sigma_p / \sigma_{obs}$$

 $\beta = (\mu_p - \mu_{obs}) / \sigma_{obs}$

Finally, the Eq.(12) was obtained by substituting Eq.(11) into Eq.(9) as follows:

$$MSE = 2\sigma_p \sigma_{obs} (1 - R) + (\sigma_p - \sigma_{obs})^2 + (\mu_p - \mu_{obs})^2$$
(12)

**Related references:**

- Murphy, A.: Skill scores based on the mean square error and their relationships to the correlation coefficient, Monthly Weather Review, 116, 2417–2424, 1988.
- Nash, J.E., Sutcliffe, J.V.: River flow forecasting through. Part I. A conceptual models discussion of principles, Journal of Hydrology, 10, 282–290, 1970.

8. Fig. 2, the spatial pattern for ERA-Interim looks pretty different from that for ERA5 and others, especially across the arid northwest and regions along the coasts. Please doublecheck it, otherwise, give an explanation.

**Response**: We doublechecked Figure 2, and found that the figure legends of NCEP/DOE R2 and ERA-Interim were wrong. The large difference between ERAI and ERA5 in the regions along the coasts attributes to the spatial resolution (Line 725).

9. Line 195, the larger rRMSEs in the Yangtze-Huai basin may be associated with the irrigation influence on the in situ observations. However, it's hard to think of its direct links to monsoon precipitation.

**Response**: Thanks for your advice. In order to remove the bias error caused by the mismatch of spatial representativeness between in situ data and all SM products, the ubRMSE was introduced instead of relative RMSE to evaluate temporal dynamic variability as showed in Figure 3. The results were showed as follows:

Line 246-251: The distribution of the ubRMSE for all stations is shown in Fig. 3 to evaluate temporal SM dynamical variability. By removing the bias, the NCEP product has the lowest ubRMSE with values between 0.01 and 0.03 m3/m3, indicating its better performance at capturing the temporal variation of in situ SM. Large ubRMSE are found for the ESA CCI with values large than 0.04 m3/m3, indicating that this remote sensing product needs to be improved at temporal variation. Spatially large ubRMSE are also found in the Yangtze-Huai region and in the south of Northeast China, which may be attributed to the high SM values. A possible explanation for poor performance in the NC region might be that this region is strongly influenced by irrigation. 10. Fig. 4, the regionally averaged observations show higher soil moisture in NW than the other three regions. It is NOT consistent with the precipitation patterns in Fig. 1. The discrepancy should be discussed a little bit more.

**Response:** Generally, the northwest region was located in the semi-arid region of China, where the annual mean precipitation between 200-400 mm. The NW region selected in this study located in the east of the Northwest China, where is the transitional zone from semi-humid region to semi-arid regions. As showed in Figure 1, the NW region consists of the arid, semi-arid, and semi-humid regions. Besides, soil moisture is influenced not only by precipitation, but also by the evaporation (affected by temperature and wind speed, etc), soil types, and other related factors. The JJA soil moisture value is obviously influenced by the contribution of both precipitation and evaporation, and we have added more discussion in section 3.4.

Line 359-367: Previous studies have showed that soil moisture is influenced by the combination of precipitation and evaporation, in which land surface evaporation is linked with temperature and surface net radiation (Jasper et al., 2006; Harmsen et al., 2009). Figure 12 shows scatter plots of (a, d, g) precipitation, (b, e, h) temperature, and (c, f, i) net radiation anomalies versus observed SM anomalies over different regions in (left column) MAM, (middle column) JJA, and (right column) SON seasons. Obvious positive correlations are found between precipitation and SM in the YH regions during MAM and SON seasons, and in the NE and NC regions during JJA season. Temperature and net radiation show negative correlation with in the NE, NC, and YH regions. The correlation coefficient is low for all meteorological variables in the NW region, which may be attributed to the special soil type there. Soil moisture in the NE and NC regions tends to be influenced by temperature during cold seasons. SM in the YH region tend to be influenced by radiation during warm seasons, due to the large evaporation there.

11. 3.2.2 Seasonality, since the previous results talk about the summer (JJA) soil moisture comparisons with observations, how the seasonal soil moisture were selected in this section should be clarified further. Further, the soil moisture discussed in the manuscript focused on the top soil layer (0-10 cm), so I guess its seasonality connected closely to precipitation annual cycle. However, in Fig. 6, it looks not so, please discuss it further.

Response: Besides the JJA SM, we also calculate monthly SM from 1981-2013 during

unfrozen and snow-free seasons to study the seasonal variation of SM. The above information has been clarified in the discussion section (Section 3.5).

Line 359-367: Previous studies have showed that soil moisture is influenced by the combination of precipitation and evaporation, in which land surface evaporation is linked with temperature and surface net radiation (Jasper et al., 2006; Harmsen et al., 2009). Figure 12 shows scatter plots of (a, d, g) precipitation, (b, e, h) temperature, and (c, f, i) net radiation anomalies versus observed SM anomalies over different regions in (left column) MAM, (middle column) JJA, and (right column) SON seasons. Obvious positive correlations are found between precipitation and SM in the YH regions during MAM and SON seasons, and in the NE and NC regions during JJA season. Temperature and net radiation show negative correlation with in the NE, NC, and YH regions. The correlation coefficient is low for all meteorological variables in the NE and NC regions tends to be influenced by temperature during cold seasons. SM in the YH region tend to be influenced by radiation during warm seasons, due to the large evaporation there.

**Related references:**

- Jasper K, Calanca P, Fuhrer J. Changes in summertime soil water patterns in complex terrain due to climatic change. Journal of Hydrology, 2006, 327: 550-563.
- Harmsen E W, Norman L M, Nicole J S, J E Gonzalez. Seasonal climate change impacts on evapotranspiration, precipitation deficit and crop yield in Puerto Rico. Agricultural Water Management, 2009, 96: 1085-1095.

12. Line 230, 'snow or frozen soil during these periods.' The frozen seasons should be excluded in the comparisons, otherwise the model soil moisture is virtually a different variable from the observed.

**Response**: In the former manuscript, we only discarded in situ soil moisture data during snow or frozen days. During this revision, the months with large percent of frozen and snow days were discarded for comparison. Furthermore, if the in situ observation were missing, all reanalysis data at the same period were also treated as missing value.

Line 156-159: Considering that the field capacity and the dry bulk density are not measured at all stations, data from 119 stations are selected from 1981 to 2013. Not all in situ data were suitable for evaluation given instrumental error and observational

conditions, for example, the available measurement period, installation depth and sensor placement. Therefore the evaluation was conducted in unfrozen and snow-free seasons, such as June-July-August (JJA).

Line 227-231: The comparisons were performed as follows: (i) make a correspondence between all soil moisture data sets and in situ SM by using the values at the nearest neighbor grids; (ii) compare all the SM products at regional scales by calculating the reginal average of monthly value of all SM products, which has been proved can reduce the uncertainty caused by grid mismatch to some extent (Nie et al., 2008); (iii) if the in situ observation were missing, all reanalysis data at the same period were also treated as missing value, which were not taking into account.

**Related references:**

- Nie, S., Luo, Y., Zhu, J.: Trends and scales of observed soil moisture variation in China, Advance in Atmosphere Science, 25, 43–58, 2008.
- 13. Line 286, 'The SC-PDSI is utilized (Wells et al., 2004).', for what is SC-PDSI used?

**Response**: The SC-PDSI is utilized here to define different dry/wet conditions. To be simplicity, this sentence and the following one has been combined into one sentence as: **Line 344**: *Figure 10 shows the rBias under different humid/arid conditions by utilizing SC-PDSI (Wells et al., 2004)*

**Related references:**

Wells, N., Goddard, S., and Hayes, M. J.: A self-calibrating palmer drought severity index, J. Clim., 17, 2335–2351, https://doi.org/10.1175/1520-0442, 2004.

**Referee Comments#2:**

**General comments:**

This interesting analysis used in-situ observations in China to evaluate several reanalysis- and RS-based SM products. While it is a nice self-contained study with seemingly comprehensive analyses, I found the study lacking sufficient physical explanations supporting several findings of their analyses. Also, some figures are not very well presented and need to be updated. Therefore, I'd suggest the authors go through moderate revisions before this paper can be publishable. Below are some suggestions to improve the paper:

**Response:** Thanks very much for your constructive suggestion. During this revision, the manuscript is improved by focusing on the following issues:

- (1) Evaluation strategies have been improved by (i) using unbiased root mean square error (ubRMSE) to remove the bias error caused by the mismatch of spatial representativeness between in situ data and all SM products; (ii) removing all the product data (including remotely sensed and reanalysis) when in situ observation were missing. As a result, all the related figures have been refined and corrected.
- (2) More physical explanations have been added in the Results and Discussion Section.

At the discussion section, uncertainties caused by comparing in situ observations with all products at different layers and grid mismatch are discussed.

Line 369-374: ESA CCI SM product showed the top layer soil content at 5-cm depth or so. The in-situ measurement depth and model output are at the depth of 0-10cm, which were also treated as the top layer soil content. Such difference would also cause representativeness errors. Previous studies have found that there is a close relationship between surface SM and SM in the upper ten centimetres (i.e., Albergel et al., 2008; Dorigo et al., 2015), so the SM measurements at the depth of 10 cm were chosen as the reference to evaluate satellite-based and reanalysis products. Furthermore, introducing ubRMSE and conducting comparison at regional scale can remove the bias error caused by mismatch of grid cell to some extent.

We further discuss why ESA CCI showed lower correlation with in situ observations.

Line 375-382: The ESA CCI combined data generally increase the number of

observations available for a time period but the correlation coefficients were not better than those of the best performing single dataset (Dorigo et al., 2015). Dorigo et al. also studied the possible reasons of input data, and found that the low correlation of combined product possibly due to the merging procedure, including the influence of vegetation (Taylor et al., 2012), the different original overpass time, and the scaling of high resolution ASCAT product to lower resolution reference products. Beck et al. (2021) found that ESA CCI SM performed better in eastern Europe in terms of high-frequency fluctuations, and speculated the overall performance of ESA CCI may be not so good due to incorporating ASCAT that performed less well.

The physical explanations of spatio-temporal SM variation have also been added.

Line 389-394: Precipitation and evaporation are found to be the most important determinant of soil moisture simulation performance, in which the evaporation is associated with temperature and radiation (Gottschalck et al., 2005; Mall et al., 2006; Chen & Yuan, 2020). SM value in the analysis is overestimated, partly due to the reason that the JJA precipitation over China is overestimated by models (e.g., Luo et al., 2013; Yun et al., 2020). The largest bias of precipitation over the Tibetan and Yun-Gui Plateaus, the North China Plain, and the southern mountains, which gives one the explanation why reanalysis products represent the worst performance over the NC region.

The detail about soil type and texture has been added.

Line 395-404: Soil type and soil texture are also important elements for soil moisture estimation. In the southwest of the NE region, the sand fraction of the topsoil can reach about 80%-90%, and the sand fraction and clay fraction of the topsoil are around 30%-40% and 10%-30% respectively (Shangguan et al., 2012) in the north NE region. The inconsistent of the soil types over the NE region might interpret why the large inconsistency of spatial distribution were found. In the northwest of the NW region, sand fraction is larger than 80%, and the sand fraction is low in the southeast of the NW region. The large difference of soil types over the north NW region is one of the reasons that all products show poor performance. In the NC and YH regions, sand and clay fraction of the topsoil account for about 10%-20% and 30%-50%, 30%-50% and 0-20% respectively. The different performance over the NC and YH regions gives hints that reanalysis products tend to performance worse when the soil contains large

percentage of sand because of its high porosity and low water holding capacity.

**Insufficient explanations/supports:**

1: ESA-CCI seems to not represent seasonality well. Why? It seems no variation there. I think this explanation on "which may be because of snow or frozen soil during these periods" is too thin. To me this still does not explain well on why worst seasonality are there.

**Response**: In the revised manuscript, monthly SM data in cold seasons (frozen and snowing) were deleted, so we deleted this sentence.

**Line 285-286:** *ESA CCI yields the worst seasonal cycle results considering the changing tendency, which may be because of lack of available data by conditional constraints of satellite sensors.*

We also add the explanation in the Discussion section.

Line 375-381: The ESA CCI combined data generally increase the number of observations available for a time period but the correlation coefficients were not better than those of the best performing single dataset (Dorigo et al., 2015). Dorigo et al. also studied the possible reasons of input data, and found that the low correlation of combined product possibly due to the merging procedure, including the influence of vegetation (Taylor et al., 2012), the different original overpass time, and the scaling of high resolution ASCAT product to lower resolution reference products. Beck et al. (2021) found that ESA CCI SM performed better in eastern Europe in terms of high-frequency fluctuations, and speculated the overall performance of ESA CCI may be not so good due to incorporating ASCAT that performed less well.

2: it seems discouraging that none of the products available captures the anomalies well especially in NC. Can the author provide some feasible explanations on why this is the case, and discuss how this could influence applications in those regions and what are the potential future directions for improvements?

**Response:** What is worthy to say is that, Figure 8 showed the interannual anomalies of JJA SM. Surface SM is a variable associate with precipitation and evaporation, both of which fluctuate greatly with time in the JJA seasons. To improve the quality of SM, all

reanalysis data would improve their performance in representing precipitation and evaporation, especially during extreme events.

We also added some discussion in Line392-394:

The largest bias of precipitation overestimation using the hourly 31-km-resolution ERA5 reanalysis data is found over the Tibetan and Yun-Gui Plateaus, the North China Plain, and the southern mountains, which gives one the explanation why reanalysis products represent the worst performance over the NC region.

3.Line 301: I think "which is partly due to the combined influence of longwave and shortwave radiation" does not sufficiently explain why low correlation there. Please expand what you mean exactly. Also, if separation of LW and SW radiation helps, would it be possible to use LW and SW data to re-draw this scatter plot?

**Response**: Figure 12 was improved by adding more information about the MAM and SON seasons, and the description was also refined as follows.

Line 359-367: Previous studies have showed that soil moisture is influenced by the combination of precipitation and evaporation, in which land surface evaporation is linked with temperature and surface net radiation (Jasper et al., 2006; Harmsen et al., 2009). Figure 12 shows scatter plots of (a, d, g) precipitation, (b, e, h) temperature, and (c, f, i) net radiation anomalies versus observed SM anomalies over different regions in (left column) MAM, (middle column) JJA, and (right column) SON seasons. Obvious positive correlations are found between precipitation and SM in the YH regions during MAM and SON seasons, and in the NE and NC regions during JJA season. Temperature and net radiation show negative correlation with in the NE, NC, and YH regions. The correlation coefficient is low for all meteorological variables in the NE and NC regions tends to be influenced by temperature during cold seasons. SM in the YH region tend to be influenced by radiation during warm seasons, due to the large evaporation there.

4. 12 & L298-L302: overall I think it's an interesting figure. However, authors fail to explain in more detail on the underlying physical mechanisms responsible for these correlations and why they wanted to perform these analyses. This paragraph is too thin. In addition, It seems these plots are more driven by the availability of data, instead of driven by hypothesis testing needs. It would be helpful for the authors to put more

thoughts on this figure and provide readers with more insights on why they chose to do the analysis and what's new after doing the analysis.

**Response:** The aim of this figure is to study the soil moisture memory in different seasons. We have provided more insights as follows:

Line 307-309: The information of soil moisture autocorrelation gives hint for the assimilation of surface soil moisture into land surface models (Crow and Van den Berg, 2010), in which during summer and winter, some other related meteorological elements should be considered.

**Related references:**

Crow, W., and Van den Berg, M.: An improved approach for estimating observation and model error parameters in soil moisture data assimilation, Water Resources Research, 46(12), doi:10.1029/2010WR00940, 2010.

**Figure presentation problems:**

5. It is very difficult to distinguish in-situ line in Fig. 6 as it can be confused with ERA-5. I'd suggest to use thicker black line to denote in-situ observations in Fig. 6. Also, be better to use consistent legend with Fig. 4 & Fig. 8.

**Response**: As suggested, thicker black line has been used to denote in-situ observations in Fig. 6. Furthermore, the legends in Fig. 4 and Fig.8 have been unified.

6. : I think it would be very difficult for readers to directly extract useful information from this figure, partly because of the color bar used, which makes it all red (plus there are so many panels). I'd suggest to use more continuous colors, with more contrasting from 0-1, such that differences in the correlations are better presented. Since only very few locations show negative correlations, you can cap the lower bound at 0, and just mention "limited negative correlation" in the caption. This way, 0-1 can be better contrasted (using blue to red) to support your interpretation on the figure in the main text.

**Response**: Thanks for your suggestion, and the color bar has been modified in the revised manuscript. Furthermore, more discussion has been added in the main text as answered in Response#4.

Line 307-309: The information of soil moisture autocorrelation gives hint for the 18 / 33

assimilation of surface soil moisture into land surface models (Crow and Van den Berg, 2010), in which during summer and winter, some other related meteorological elements should be considered.

**Related references:**

Crow, W., and Van den Berg, M.: An improved approach for estimating observation and model error parameters in soil moisture data assimilation, Water Resources Research, 46(12), doi:10.1029/2010WR00940, 2010.

7. 10 & Fig. 11: the caption is incomplete and misleading. It did not mention which skill metrics is plotted here. Please mention it explicitly in the caption. Also, draw a reference line on 0 such that readers know where to expect good performance.

**Response**: Fig.10 and Fig.11 were changed by showing the rBias and ubRMSE of remote sensing and reanalysis SM against in situ observations under extreme (and severe) dry or extreme (and severe) wet conditions, and the figure caption has been improved.

Line 344-352: Figure 10 shows the rBias under different humid/arid conditions by utilizing SC-PDSI (Wells et al., 2004). The rBias of JJA SM between in situ observation and remote sensing/reanalysis was calculated at each in situ grid point as the bias divided by the mean of in situ observations, and then averaged over regions. All of the reanalysis products show a lower rBias under drought condition than wet condition, indicating better performance of all products under dry conditions. The largest rBias was found for all products in the NE region, implying that the largest uncertainty would appear in this region during extreme events. Large difference of rBias between dry and wet conditions was observed in the NW region, implying that all products fail to represent the SM value when the water content is high. The largest rBias is found for ERAI under severe wet conditions in NE, with an average bias of 144.4%. The best performance is found for ESA CCI SM in NW, with averaged rBias of 10.0%, respectively.

Line 353-358: For the ubRMSE in different regions (Fig. 11), the ubRMSE of all SM products in the NE and NW regions is noticeably high. The difference of ubRMSE between different conditions are not so large as rBias, especially in the NE region. Overall the ubRMSE for all products is larger under wet conditions, while the phase is

opposite in the NW region. The averaged bias for ESA CCI under drought conditions is smaller than that under wet conditions. The largest and smallest ubRMSE are found for the ESA CCI under wet condition in the NE region and NCEP SM products under both conditions in the YH region, respectively.

**Minor:**

8. I do not think the literature review is comprehensive. Beck et al. (2021; HESS) presented a much more comprehensive study performed at the global scale. It should be included in the Introduction and discussions on relevance to your study needs to be mentioned. I disagree with the claim in L62 that no long-term SM products have been compared with ESA-CCI. Please revise accordingly.

**Response**: Thanks for your recommendation. We have added Beck et al. (2021, HESS) in the Introduction section, and include it in the discussion as the support of the physical mechanisms especially in the Discussion section.

Line 379-381: Beck et al. (2021) found that ESA CCI SM performed better in eastern Europe in terms of high-frequency fluctuations, and speculated the overall performance of ESA CCI may be not so good due to incorporating ASCAT that performed less well.

Line 412-414: Beck et al. (2021) concluded that assimilating satellite soil moisture estimate maybe not improve more than increasing model resolution or improving soil moisture simulation ability, which is in line with our results. This suggest that improving model simulation performance of SM is beneficial especially at long-term scales.

Furthermore, "no long-term SM products have been compared with ESA-CCI" has been adjusted as follows:

Line 64-65: However, few studies on long-term SM products over 30 years have been compared with the ESA CCI product using in situ measurements in East China, and thus, more in-depth evaluation needs to be done.

**Overall comment:**

9. In fact, I like the study very well because it is self-contained, with comprehensive analysis, and the writing is good too. However, I am thinking what could be more useful to the community, is perhaps for the authors to share their in-situ soil moisture observations through posting the data via figshare or other publicly accessible data portal. It seems to me that this study is only unique because of its observations, which are generally not shared with the public. If the data can be shared properly with the whole community, people may find more innovative ways of using the data for other research purposes such as drought monitoring. Is this something that the authors are considering? It could be helpful to at least comment on or discuss this issue in an academic paper.

**Response:** Thanks for your suggestion, and the in situ data can be obtained by requesting from the International Soil Moisture Network website (https://ismn.geo.tuwien.ac.at/en/), and National Meteorological Information Center of China (NMIC, http://data.cma.cn/site/index.html). We have added the above information in the Data Availability section.

**Line 433-436:** The updated Chinese soil moisture presented as volumetric soil moisture  $(\theta v, unit=m^3 m^{-3})$  for 1981 to 1999 was downloaded from the International Soil Moisture Network website (https://ismn.geo.tuwien.ac.at/en/). The in situ SM measurements are obtained by requesting from the website of National Meteorological Information Center of China (NMIC, http://data.cma.cn/site/index.html).

**Response to the Community Comments:**

**Community Comments**

1. The authors need to reappraise their motive of this study, because NOAA and NCEP soil moisture (SM) products (a spatial resolution of 2 degrees) are usually not qualified for hydro-meteorological studies (flood or drought as reviewed by Peng et al. 2020, in Remote Sensing of Environment) in mainland China. As pointed out by the other reviewer, such coarse spatial resolutions would cause representativeness errors. Although spatial averaging to some extent can alleviate such an effect, I still think errors of representativeness (together with differences in effective soil depth) might contribute substantially to the bias values. That is probably the reason why CCI (0.25 degrees) and ERA-5 (31 kilometers) have a slightly better performance.

**Response**: During this revision, the bias error caused by the mismatch of spatial representativeness between in situ data and all SM products has been removed by introducing the unbiased root mean square error (ubRMSE) (see Figure 3 (Line 730), Figure 11 (Line 760) and Table 3 (Line 775)). Furthermore, the comparison was conducted at regional scales by calculating the reginal average of monthly value for all SM products, which can reduce the uncertainty caused by grid mismatch to some extent.

---

## Author Response (AR2)

**Overall response:**

We would like to thank the editor and two anonymous reviewers for their suggestions, which are helpful to improve this manuscript. During this revision, we have considered the suggestions seriously by correcting the typos, revising figures, unifying the units, and finally improving the writing. The revisions are marked using the "Track Changes" function in the revised manuscript, and the point-by-point responses to the reviewers' comments are listed below.

**Referee Comments#1:**

**General comments:**

I appreciate the authors' efforts in addressing my concerns and making revisions. While the efforts are appreciated, I found the writing has not been polished in this revision, so the writing still needs more efforts.

**Response:** Thanks very much for your kind consideration of this manuscript. During this revision, we have considered your suggestions seriously by correcting the typos, revising figures, unifying the units, and finally improving the writing.

For example, "m$^3$/m$^3$" has been changed as "m$^3$ m$^{-3}$", and "mm/year" has been changed into "mm a$^{-1}$".

Furthermore, "~" has been changed into "-".

Below are a few examples:

1. L285-286: it is not clear what do you mean by "considering the chancing tendancy", and what do you mean by "conditional constraints of satellite sensors". Please revise the writing. It is better to consult native speakers.

**Response:** "considering the chancing tendancy" has been changed into "with respect to temporal variation", and "conditional constraints of satellite sensors" has been changed into "which may be because of large percentage of missing data".

**Line 287-288:** *ESA CCI yields the worst seasonal cycle results with respect to temporal variation, which may be because of large percentage of missing data.*

2. L374-380: change to "and found that xxx was possibly due to xxx".

**Response:** Corrected as suggested.

**Line 380-382:** *Beck et al. (2021) found that ESA CCI SM performed better in eastern Europe in terms of high-frequency fluctuations, and found that the overall performance of ESA CCI may be not so good was possibly due to the incorporation of ASCAT that performed less well.*

3. L374-308: change to "due to the incorporation of ASCAT that performed less well"

**Response:** Corrected as suggested.

**Line 380-382:** *Beck et al. (2021) found that ESA CCI SM performed better in eastern Europe in terms of high-frequency fluctuations, and found that the overall performance of ESA CCI may be not so good was possibly due to the incorporation of ASCAT that performed less well.*

4. L390-392: it is incorrect to say "which gives one the explanation", what do you mean

exactly? It needs to be revised.

**Response:** This sentence has been changed into "which gives one of the explanations".

**Line 393-395:** *The largest bias of precipitation overestimation using the hourly 31-km-resolution ERA5 reanalysis data is found over the Tibetan and Yun-Gui Plateaus, the North China Plain, and the southern mountains, which gives one of the explanations why reanalysis products represent the worst performance over the NC region.*

L385-366: change to "have shown"

**Response:** Corrected.

**Line 360-361:** *Previous studies have shown that soil moisture is influenced by the combination of precipitation and evaporation, in which land surface evaporation is linked with temperature and surface net radiation (Jasper et al., 2006; Harmsen et al., 2009).*

L401-403: I don't think "maybe not improve more than" is correct in English. Please revise.

**Response:** "maybe" has been changed into "may".

**Line 413-415:** *Beck et al. (2021) concluded that assimilating satellite soil moisture estimate may not improve more than increasing model resolution or improving soil moisture simulation ability, which is in line with our results.*

Other issues with very vague wordings:

L385-366: due to which special soil type? Please offer more details.

**Response: "special soil type" has been changed into "large fraction of sand".**

**Line 366-367:** *The correlation coefficient is low for all meteorological variables in the NW region, which may be attributed to the large fraction of sand there.*

L306-308: what "some other meteorological variables"? These words are too vague and please clarify/revise.

**Response:** The detailed information about the meteorological variables has been added.

**Line 308-310:** *The information of soil moisture autocorrelation gives hint for the assimilation of surface soil moisture into land surface models (Crow and Van den Berg, 2010), in which during summer and winter, the influence of meteorological elements (e.g., precipitation, temperature, evaporation, etc) should be considered more.*

These are just some examples captured here. I suggest the authors to thoroughly revise the writing by consulting native speakers or professional academic writing, before publishing their paper.

**Referee Comments#2:**

**General comments:**

The authors have clearly responded to all the questions raised before. Now I have no more suggestions. After correcting some typos, I am willing to support the publication of the manuscript. For example, in Fig.2 the units should be represented properly using a superscript.

**Response:** Thanks very much for your kind consideration of this manuscript. During this revision, we have considered your suggestions seriously by correcting the typos, revising figures, unifying the units, and finally improving the writing. For example, in Fig.2 the units have been represented properly using a superscript. Furthermore, some issues with very vague wordings have been resolved.

For example, "$m^3/m^3$" has been changed as "$m^3\ m^{-3}$", and "mm/year" has been changed into "$mm\ a^{-1}$".

Furthermore, "~" has been changed into "-".